# Screening and Structural Characterization of Heat Shock Response Elements (HSEs) in *Entamoeba histolytica* Promoters

**DOI:** 10.3390/ijms25021319

**Published:** 2024-01-21

**Authors:** David Dorantes-Palma, Salvador Pérez-Mora, Elisa Azuara-Liceaga, Ernesto Pérez-Rueda, David Guillermo Pérez-Ishiwara, Misael Coca-González, María Olivia Medel-Flores, Consuelo Gómez-García

**Affiliations:** 1Laboratorio de Biomedicina Molecular 1, ENMyH, Instituto Politécnico Nacional, Mexico City 07320, Mexico; pdorantes1700@alumno.ipn.mx (D.D.-P.); sperezm1510@alumno.ipn.mx (S.P.-M.); dperez@ipn.mx (D.G.P.-I.); mcocag1900@alumno.ipn.mx (M.C.-G.); molivof@ipn.mx (M.O.M.-F.); 2Posgrado en Ciencias Genómicas, Universidad Autónoma de la Ciudad de México, Mexico City 03100, Mexico; elisa.azuara@uacm.edu.mx; 3Unidad Académica del Estado de Yucatán, Instituto de Investigaciones en Matemáticas Aplicadas y en Sistemas, Universidad Nacional Autónoma de México, Mexico City 97302, Mexico; ernesto.perez@iimas.unam.mx

**Keywords:** *Entamoeba histolytica*, promoters, HSEs, EhHSTFs, in silico, genome screening, gene ontology, docking molecular, thermal shock stress, gene regulation

## Abstract

*Entamoeba histolytica* (*E. histolytica*) exhibits a remarkable capacity to respond to thermal shock stress through a sophisticated genetic regulation mechanism. This process is carried out via Heat Shock Response Elements (HSEs), which are recognized by Heat Shock Transcription Factors (EhHSTFs), enabling fine and precise control of gene expression. Our study focused on screening for HSEs in the promoters of the *E. histolytica* genome, specifically analyzing six HSEs, including Ehpgp5, EhrabB1, EhrabB4, EhrabB5, Ehmlbp, and Ehhsp100. We discovered 2578 HSEs, with 1412 in promoters of hypothetical genes and 1166 in coding genes. We observed that a single promoter could contain anywhere from one to five HSEs. Gene ontology analysis revealed the presence of HSEs in essential genes for the amoeba, including cysteine proteinases, ribosomal genes, Myb family DNA-binding proteins, and Rab GTPases, among others. Complementarily, our molecular docking analyses indicate that these HSEs are potentially recognized by EhHSTF5, EhHSTF6, and EhHSTF7 factors in their trimeric conformation. These findings suggest that *E. histolytica* has the capability to regulate a wide range of critical genes via HSE-EhHSTFs, not only for thermal stress response but also for vital functions of the parasite. This is the first comprehensive study of HSEs in the genome of *E. histolytica*, significantly contributing to the understanding of its genetic regulation and highlighting the complexity and precision of this mechanism in the parasite’s survival.

## 1. Introduction

Amoebiasis is an acute or chronic gastrointestinal disease caused by the protozoan parasite, *E. histolytica*. This disease has two major clinical manifestations: hemorrhagic colitis and liver abscesses [1]. It is estimated that approximately 50 million people worldwide are infected with this parasite, making amoebiasis a serious global health problem [2]. Infection is caused by water or food contaminated with feces containing amoebic cysts [3]. *E. histolytica* has a two-stage life cycle: the infective stage in the environment is the cyst, and the motile stage within the host is the trophozoite. Infection with this parasite is endemic in many parts of the world, primarily in areas where sanitation infrastructure is inadequate [4,5].

When *E. histolytica* infects humans, it encounters diverse environments, each prompting distinct changes in its gene expression, enabling rapid adaptation to each specific microenvironment. A prominent example is the response to heat stress, where a sudden rise in temperature triggers the synthesis of heat shock proteins (HSPs). This response represents a conserved defensive mechanism that organisms have evolved to protect themselves against acute exposure to elevated temperatures [6,7].

The expression of genes encoding heat shock proteins (HSPs) is regulated by a master transcription factor known as the heat shock transcription factor (HSF). HSFs possess a remarkably intricate and complex activation mechanism [8]. In response to heat stress, HSF, located in the cytoplasm, undergoes phosphorylation, trimerization, and subsequent translocation to the nucleus. There, through its DNA-binding domain (DBD), HSF specifically recognizes the heat shock response element (HSE) in the promoters of genes, such as HSPs, and activates their transcription [9]. Currently, there is a notable variation in the number of HSFs identified across different species in the evolutionary spectrum. These counts range from a single gene encoding HSF to more than fifty. For instance, *Saccharomyces cerevisiae* (*S. cerevisiae*, baker’s yeast), *Drosophila melanogaster* (*D. melanogaster*, fruit fly), and *Caenorhabditis elegans* (*C. elegans*, nematode worm) each have one gene encoding for HSF [10]. In contrast, *Homo sapiens* (*H. sapiens*, human) possesses six, *Arabidopsis thaliana* (*A. thaliana*, thale cress) has twenty-one [10], *Solanum lycopersicum* (*S. lycopersicum*, tomato) has twenty-four [10,11], *Glycine max* (*G. max*, soybean) has a significantly higher count with fifty-three HSF genes [12], and in *Triticum aestivum* (*T. aestivum*, wheat) at least fifty-six genes for HSF have been reported [13]. These HSFs are involved in diverse stress responses, including thermal stress and drug exposure, and play roles in neurodegenerative diseases like Alzheimer’s and Parkinson’s, cancer, and various metabolic processes, including cell division and development [14,15,16]. The HSE sequences typically consist of at least three contiguous 5-base pair inverted repeats (motifs): nGAAn or AGAAn (nTTCnnGAAnnTTCn), located in the promoter regions of *hsp* genes [17,18,19] and other genes like HMOX-1 [20] and IL-6 in *H. sapiens* [21], *Apx1* in *A. thaliana* [22], *cup1* (Metallothionein) in *S. cerevisiae*, [23] and HSP70 in *D. melanogaster* [24]. However, the composition and arrangement of these motifs can vary among different organisms.

In *E. histolytica*, the response to thermal stress has been extensively studied, and it has been established that this parasite effectively manages such stress, primarily through the expression of HSPs [25,26]. The parasite possesses a family of seven EhHSTFs. These are registered in the AmobaDB database (https://amoebadb.org/amoeba/app, accessed on 9 December 2022) under the identifiers EHI_008230 for EhHSTF1, EHI_049510 for EhHSTF2, EHI_087630 for EhHSTF3, EHI_142120 for EhHSTF4, EHI_137000 for EhHSTF5, EHI_008660 for EhHSTF6, and EHI_200020 for EhHSTF7. In a recent study, Bello et al. [27] demonstrated in vitro that the factor EhHSTF7 specifically identifies the HSE located in the promoter of the *Ehpgp5* gene, which is associated with multidrug resistance in *E. histolytica*. Research conducted on trophozoites of the A HM-1:IMSS strain of *E. histolytica*, exposed to 8 µM of emetine, an amoebicidal agent, for 24 h, revealed that selective inhibition of the *Ehhstf7* gene through siRNA leads to a significant reduction in the expression of *EhPgp5*. These findings suggest that EhHSTF7 plays a crucial role in the multidrug resistance of *E. histolytica* [27,28].

In addition, some HSEs have been identified in *E. histolytica* gene promoters, such as Ehpgp5 at position −150/−137 (5′-ataGAAatttTTCata-3′); Ehhsp100 at position −223/−229 (5′-aagGAActtGAAgaa-3′); EhMLBP (methyl-binding protein) at position −33/−2 (5′-agaaGAActaataactGAAaattataatTTC-3′); and three HSEs in the EhrabB gene coding for a Rab GTPase, named EhrabB4 at position −577/−543 (5′-gTTCttcagtaagaGAA-3′), EhrabB1 at position −15/−2 (5′-cTTCaTTCcatggtttTTC-3′), and EhrabB5 at position 624/−608 (5′-aTTCtacaaaaaGAA-3′) [28,29,30,31].

Moreover, the regulatory influence of HSF binding to an HSE extends beyond genes encoding HSPs. This mechanism also governs the expression of additional genes, exemplified by interleukins in *H. sapiens* [32]. Consequently, HSFs and HSEs are pivotal in the transcriptional regulation of genes implicated in several cellular processes, encompassing oxidative stress; exposure to heavy metals, toxins, and pharmacological agents; and alterations in pH, among others. These processes are predominantly associated with responses to environmental stressors or cellular changes [33,34].

Given that *E. histolytica* possesses several EhHSTFs and HSEs that are critical for controlling key processes that contribute to its survival in stressful host environments and recognizing the current lack of detailed mapping of HSEs in its genome, including their characterization and the genes that may be regulated by them, we decided to conduct an exhaustive investigation and structural analysis of HSEs within the parasite genome.

## 2. Results

### 2.1. Screening HSEs in Promoter Regions of the E. histolytica Genome

The HSEs Ehpgp5 (5′-ataGAAatttTTCata-3′), Ehhsp100 (5′-aagGAActtGAAgaa-3′), EhrabB4 (5′-gTTCttcagtaagaGAA-3′), EhrabB1 (5′-cTTCaTTCcatggtttTTC-3′), EhrabB5 (5′-aTTCtacaaaaaGAA-3′), and Ehmlbp (5′-agaaGAActaataactGAAaattataatTTC-3′) were systematically searched within the promoter regions (−500 to +50 bp) of the 8343 genes present in the *E. histolytica* genome. A total of 2578 HSEs were identified, with 1412 HSEs located in the promoter regions of 1010 hypothetical genes, and 1166 HSEs in the promoter regions of 957 coding genes (Figure 1a). This result indicates that approximately 24% of *E. histolytica* genes may be regulated by an HSE (Figure 1b).

Among the six HSEs analyzed, Ehhsp100 was the most abundant in the promoters, with a total of 702 HSEs (377 in hypothetical genes and 325 in coding genes), followed by Ehpgp5 with 645 HSEs (341 in hypothetical genes and 304 in coding genes). Ehmlbp accounted for 531 HSEs (319 in hypothetical genes and 212 in coding genes), EhrabB1 for 303 (155 in hypothetical genes and 148 in coding genes), EhrabB5 for 234 (136 in hypothetical genes and 98 in coding genes), and EhrabB4 for 163 HSEs (84 in hypothetical genes and 79 in coding genes) (Table 1).

Due to the unknown identities and functions of the proteins encoded by hypothetical genes in *E. histolytica*, we continued the analysis by focusing solely on the presence of HSEs in the promoters of coding genes. Interestingly, we found that the total number of genes, 957, was lower than the number of HSEs identified, which was 1166. This discrepancy could be attributed to the possibility that some genes have multiple occurrences of the same HSE in their promoter regions, or they might exhibit combinations of different HSEs (Figure 2a).

To determine the frequency of each of the six HSEs, either individually or in combination, within the amoeba promoters, we constructed a Venn diagram (Figure 2b). This diagram facilitated visualization of both the overlapping and distinct occurrences of these HSEs across the gene promoters, offering valuable insights into the distribution of these regulatory elements within the amoeba’s genome. Our analysis revealed that out of the 1166 HSEs, 787 genes possess a unique HSE. Specifically, 214 genes contain the Ehpgp5 element, 199 genes possess the Ehhsp100 element, 144 genes carry the Ehmlbp element, 109 genes feature the EhrabB1 element, 68 genes include the EhrabB5 element, and 53 genes have the EhrabB4 element (Figure 2a, Table 2, Appendix A). While 138 genes have two HSEs, 43 genes exhibit the same HSE combination, and 96 genes have two different HSEs (Appendix A). The most frequent combination is Ehhsp100-Ehpgp5, present in 22 genes, including a helicase (EHI_184530), an EhHSTF (EHI_184530), and the chaperone clpB (EHI_155060). The second most common combination is Ehhsp100-Ehmlbp, found in 15 genes, which includes the DNA polymerase gene (EHI_132860), lipase (EHI_099800), and a protein kinase (EHI_035500). Furthermore, twenty-six genes have three HSEs, with six genes displaying the same combination, and twenty genes having three different HSEs (Appendix A). For instance, the enolase gene (EHI_130700) and the inositol polyphosphate kinase gene (EHI_025390) both have the combination Ehhsp100-Ehmlbp-Ehpgp5. On the other hand, the genes encoding the RhoGAP domain-containing protein (EHI_114370) and a subunit of DNA primase (EHI_147530) exhibit the combination Ehhsp100-Ehpgp5-EhrabB5. The other four genes specifically had at least one EhrabB-type HSE in their promoters. For example, the alpha importin gene (EHI_148070) exhibited the elements Ehhsp100-Ehpgp5-EhrabB4; the actin gene (EHI_182900) contained the HSEs Ehhsp100-Ehmlbp-EhrabB4; the histone 3 gene (EHI_193360) carried the sequences EhrabB1-EhrabB4-EhrabB5, and the gene encoding the nuclear complex protein 4 had the sequences Ehmlbp-EhrabB1-EhrabB5 (Figure 2a, Table 2, Appendix A).

Only three genes were identified with four HSEs. The gene for the vesicle-associated protein (EHI_081370), which plays a role in endocytosis and exocytosis, displayed four identical HSEs (Ehhsp100). Furthermore, two genes harbored four distinct HSEs. The gene encoding the protein with the IBR domain (EHI_120320) had one Ehpgp5 element and three Ehhsp100 elements (Figure 2a, Table 2, Appendix A), and the gene for the phenylalanyl-tRNA synthetase beta subunit (EHI_002130) contained three Ehpgp5 elements and one Ehhsp100 element. Remarkably, only two genes exhibited five HSEs. The gene encoding the DEAD/DEAH box helicase (EHI_169630), crucial in RNA-related molecular processes from transcriptional regulation to mRNA degradation, had five Ehhsp100 HSEs. The gene for the cell division cycle protein 123 homolog (EHI_128440), involved in cell cycle regulation, possessed one Ehpgp5 and four Ehhsp100 elements.

### 2.2. Composition of HSEs

The HSEs are typically characterized by containing two or three motifs of the GAA and/or TTC types. To characterize the nucleotide composition and organization of each HSE (Ehhsp100, Ehpgp5, Ehmlbp, EhrabB1, EhrabB4, and EhrabB5), weight matrices were constructed for each of the six elements (Appendix A). Additionally, a WebLogo was created based on the total number of HSEs identified for each element. This analysis incorporated HSEs found in the promoter regions of both hypothetical and coding genes, thereby providing a solid basis for the identification of each consensus sequence.

The results highlight specific characteristics of the HSE sequences in *E. histolytica*. The Ehpgp5 sequence, 16 nucleotides long, exhibits two highly conserved motifs: GAA at position 4 and TTC at position 11, with a four-nucleotide gap between them (Figure 3a). The Ehhsp100 sequence, consisting of 15 nucleotides, was previously reported to have two GAA motifs but is now observed to contain three, located at positions 4, 10, and 13, with only three nucleotides separating the first two (Figure 3b). The Ehmlbp element, the longest at thirty-one nucleotides, encompasses three motifs: two GAAs at positions 5 and 17, and a TTC at position 29, each separated by nine nucleotides (Figure 3f). In the EhrabB series, EhrabB1, with nineteen nucleotides, comprises three TTC motifs at positions 2, 6, and 17, separated by one nucleotide between the first two and eight between the second and third (Figure 3d). The EhrabB4 sequence (17 nucleotides) includes a TTC motif at position 2 and a GAA at position 15, with a 10-nucleotide gap (Figure 3c). Lastly, the 15-nucleotide EhrabB5 sequence features TTC and GAA motifs at positions 2 and 13, respectively, separated by eight nucleotides (Figure 3e). These findings underscore the conservation of GAA and TTC motifs in the HSEs of *E. histolytica*.

### 2.3. Location of HSEs in the Promoter

In transcriptional control, the precise location of consensus sequences is crucial for the recognition of transcription factors. Consequently, in our study, we identified the position of each of the 1166 HSE sequences in the 957 promoters of the coding genes of *E. histolytica*, shown by a density plot (Figure 3g). This analysis provided valuable information on the distribution and frequency of HSEs in the promoter regions, which helps to understand their possible regulatory functions of gene expression. Upon localizing the HSEs along the promoter region from −500 to +50 bp, we did not find any HSEs downstream of the translation initiation codon (ATG). The first HSE was identified from −1 bp of the ATG. With this information, we observed that 216 HSEs (19%) were located from −1 to −100 bp, 307 elements (26%) from −101 to −200 bp, 281 elements (24%) from −201 to −300 bp, 234 elements (20%) from −301 to −400 bp, and 128 elements (11%) from −401 to −500 bp. These results show that the majority of HSE sequences (n = 587/50.3%) are located between nucleotides −101 to −300 bp (Figure 3g, Appendix A).

The Ehhsp100 element, with 157 occurrences, was predominantly located between −100 and −300 bp in its gene promoters. Similarly, the Ehpgp5 element, with 156 instances, also showed a frequent presence in the same positional range. Although the EhrabB4 element had the fewest HSEs identified in our study (79 sequences), a significant portion of them (n = 45) were concentrated between −150 and −350 bp, as depicted in the density plot (Figure 3g). This distinct distribution of HSEs within the promoter regions may suggest their crucial role in modulating the magnitude of response and in the regulation of gene expression.

### 2.4. Identification of Core Promoter Sequence Elements in Promoters with HSEs

Subsequently, another important part of our study was to investigate whether genes with HSEs also contain other conserved regulatory elements known in *E. histolytica*, such as the non-canonical TATA box (TATTTAAA), the Inr sequence (AAAAATTCAA), and the GAAC sequence (GAACT), all of which are critical in transcriptional regulation [35]. For this purpose, we meticulously analyzed the promoter regions of genes containing HSEs to determine the presence of these additional regulatory elements. The identification of these elements, together with HSEs, could shed light on the intricate regulatory mechanisms controlling gene expression in *E. histolytica*.

The analysis showed that among the 957 genes containing HSEs, 343 (36.4%) also harbored at least one conserved regulatory sequence (TATA box, Inr sequence, or GAAC sequence) (Table 3, Appendix A). These genes exhibited one to three HSEs in their promoter regions, suggesting that a substantial proportion of genes with HSEs are co-regulated by other conserved elements. This co-regulation could be crucial in coordinating transcriptional responses to diverse cellular conditions or stimuli. Notably, the GAAC element, exclusive to the amoeba, was the most common regulatory sequence, identified in 298 genes as a singular occurrence, and was most frequently associated with the Ehhsp100 element in 104 genes (Appendix A). Examples include genes like the Rab family GTPase (EHI_003020), 40S ribosomal protein S6 (EHI_000590), and Serine acetyltransferase 1 (EHI_021570). Additionally, 45 genes possessed two HSEs, with Ehpgp5-Ehhsp100 being the most prevalent combination. Representative genes with this combination are the cell division cycle protein 123 homolog (EHI_128440) and syntaxin (EHI_139030) (Appendix A). In addition, only one gene exhibited a combination of three HSEs along with the GAAC regulatory element enolase (EHI_130700) with the sequence Ehpgp5-Ehhsp100-Ehmlbp-GAAC (Appendix A). The TATA box was identified in 68 genes, with 56 of these containing a single HSE (Appendix A). The HSE most associated with the TATA box was Ehhsp100, found in genes such as thioredoxin (EHI_062790) and the leucine-rich repeat protein of the BspA family (EHI_105370). Ten genes had dual HSEs; for example, the heat shock protein 101 (EHI_076480) with Ehpgp5-Ehhsp100 and the heat shock protein 90 (EHI_196940) with Ehhsp100-Ehmlbp. Additionally, only two genes presented a combination of three HSEs along with both a TATA box and GAAC sequence: inositol polyphosphate kinase (EHI_025390) with Ehpgp5-Ehhsp100-Ehmlbp and histone H3 (EHI_193360) with EhrabB4-EhrabB1-EhrabB5 (Appendix A). The Inr sequence was identified in only three genes: the myb-like DNA-binding domain-containing protein (EHI_009820), CXXC-rich protein (EHI_093710), and protein kinase (EHI_197540), each harboring a single HSE: EhrabB5, EhrabB1, and EhrabB4, respectively (Appendix A). This indicates that while the TATA box and GAAC element are more prevalent in the promoter regions of genes with HSEs, the Inr sequence appears to be less common, occurring in specific genes within *E. histolytica*. Additionally, 17 genes were found to possess both a TATA box and the GAAC sequence, each with one HSE. In this group, Ehhsp100 was the most frequently occurring element, present in genes such as the serine/threonine-protein phosphatase 2B catalytic subunit (EHI_109570) and Ran family GTPase (EHI_148190) (Appendix A). Six genes displaying both the TATA box and GAAC sequence had two HSEs, with the Ehpgp5-Ehhsp100 combination. These genes encode heat shock proteins (EHI_013550, EHI_017350, EHI_022620, EHI_034710, EHI_076480, and EHI_156560) (Appendix A). In our analysis, only one gene was identified in combination with the TATA/Inr regulatory elements. Conversely, none of the genes with HSEs contained all three regulatory elements (Inr, TATA box, and GAAC sequence). These findings suggest that certain combinations of regulatory elements might have distinct functional roles in gene expression regulation in *E. histolytica*.

### 2.5. Ontology Analysis of E. histolytica Genes with HSEs in Promoter Regions

Upon conducting gene ontology analysis of the 957 genes with HSEs using the PANTHER platform, the genes were categorized into three main groups: molecular functions encompassing 470 genes, biological processes including 454 genes, and protein classes comprising 731 genes (Figure 4, Appendix A). Notably, genes containing Ehpgp5, Ehhsp100, and Ehmlbp elements in their promoters were predominantly represented in all three ontological categories (Appendix A). This observation is consistent with the fact that these sequences are the most abundant in the promoters of *E. histolytica* genes (Appendix A). Additionally, within each ontological group defined by the server, genes exhibited either a unique HSE or combinations of the six HSE sequences analyzed.

### 2.6. Molecular Functions

In the molecular functions group, genes with HSEs were observed to cover 10 distinct categories. The category of catalytic activity (GO:0003824) encompasses the highest number of genes with HSEs, a total of 292, followed by the binding category (GO:0005488) with 250 genes, the Molecular Function Regulator category (GO:0098772) with 48 genes, and the ATP-dependent activity category (GO:0140657) with 42 genes (Figure 5a). Predominantly, the Ehpgp5, Ehhsp100, and Ehmlbp elements were identified as unique elements in these categories, along with combinations such as Ehpgp5-Ehhsp100 and Ehhsp100-Ehmlbp (Appendix A). These categories include genes integral to the amoeba’s vital functions, encompassing the cell cycle, trophozoite structure, transport, DNA repair, and energy production. Specific genes such as the Tyrosyl-DNA phosphodiesterase (EHI_148390) and DNA repair protein RAD51C (EHI_122860) are classified under the catalytic activity category. The binding category includes the Rho family GTPase (EHI_067390) and ATP-binding cassette protein (EHI_047760). The RhoGAP domain-containing protein (EHI_153150) and Rab GTPase-activating protein (EHI_149900) are part of the Molecular Function Regulator category. Finally, the ATP-dependent activity category features genes like the Phospholipid-transporting ATPase (EHI_141350) and AAA family ATPase (EHI_130980) (Appendix A).

Additionally, genes with HSEs are involved in diverse functional categories, such as Cytoskeletal Motor Activity (GO:0003774) with three genes, and Transporter Activity (GO:0005215) with sixteen genes. Examples include the Kinesin-like protein (EHI_140230) in the former category and the Phospholipid-transporting ATPase (EHI_141350) in the latter. Moreover, genes in the Transcription Regulator Activity (GO:0140110) and Translation Regulator Activity (GO:0045182) categories, with five and six genes, respectively, exhibit regulatory functions in essential mechanisms. Notable representatives are the Myb-like DNA-binding domain-containing protein (EHI_130710) for transcription and the Eukaryotic Translation Initiation Factor 4E (EHI_159690) for translation. The distribution of these genes across various molecular function categories, according to each of the six different HSEs, can be observed in Appendix A.

Given the fundamental importance of catalytic activity (GO:0003824) in cellular life, and its status as the category with the highest number of genes with HSEs, we delved deeper into its subcategories. Using PANTHER, we identified nine subcategories, with Hydrolase Activity (GO:0016787) being the most represented, with 176 genes from all six types of HSEs identified in *E. histolytica* promoters, predominantly Ehpgp5. Key genes in this subcategory include cysteine protease (EHI_050570) and Rab GTPase (EHI_068360), crucial for cellular transport and signaling. Following this is the Catalytic Activity Acting on a Protein subcategory (GO:01400960) with 101 genes, including notable proteins like histolysin (EHI_033710), which is involved in tissue invasion; GPI-anchor transamidase (EHI_092280) with cysteine endopeptidase activity; and the serine endopeptidase, peptidase family protein S54 (Rhomboid) (EHI_029220). Additionally, protein kinase (EHI_087600), vital for catalyzing ATP reactions, is also part of this subcategory. The third significant subcategory, Transferase Activity (GO:0016740), consists of 91 genes, mainly featuring Ehpgp5, Ehhsp100, and Ehmlbp elements, either individually or in combination. Prominent in this group are genes such as acetyltransferase (EHI_096770), Acetyltransferase GNAT family (EHI_137930), and Serine O-acetyltransferase (EHI_021570), all involved in acetyl group transfer, and Vacuolar protein sorting 26A (EHI_162540), which plays a role in protein folding (Appendix A). These findings suggest that a significant number of essential enzymes in *E. histolytica*, required to efficiently carry out chemical reactions at specific times for various metabolic pathways, could be regulated by HSEs.

### 2.7. Biological Processes

Within the biological processes functional group, there are 20 categories, but our study found that genes with HSEs were involved in only 12 of these (Figure 5b). The most populated category is Cellular Processes (GO:0009987), containing 404 genes with each of the six HSEs, accounting for 28 to 45% of the total. This is followed by Metabolic Processes (GO:0008152) with 23 to 33% (*n* = 258), Biological Regulation (GO:0065007) with 11 to 16% (*n* = 142), and Response to Stimuli (GO:0050896) with 9 to 16% (*n* = 97) (Appendix A). Representative genes in these categories include Rho family GTPase (EHI_046630) and PH domain-containing protein (EHI_091510) in Cellular Processes; phosphopyruvate hydratase (EHI_130700) and ribosomal RNA methyltransferase (EHI_098730) in Metabolic Processes; adenylyl cyclase-associated protein (EHI_136150) and peptidyl-prolyl cis-trans isomerase (EHI_054760) in Biological Regulation; and heat shock protein (EHI_022620) and heat shock protein 70 (EHI_188610) in Response to Stimulus.

In Appendix A, the distribution of genes across twelve different biological process categories is depicted, according to each of the six distinct HSEs. Within the Cellular Processes category (GO:0009987), which showcases the highest number of genes with HSE elements, there are 25 subcategories. Particularly noteworthy is the Cellular Metabolic Process subcategory (GO:0044237), encompassing the largest group of genes with HSEs (*n* = 249). Predominant within this group are the Ehpgp5, Ehhsp100, and Ehmlbp elements. Notable examples include Cell Division Protein Kinase (EHI_021680), a crucial regulator in cell cycle progression and differentiation, as evidenced in research on amoebas and other organisms [36]. Additionally, the category includes several genes for the DNA Repair Protein RAD51C (EHI_122860), which is instrumental in efficient DNA damage signaling through its role in regulating phosphorylation [37]. The subsequent subcategories with a notable presence of genes containing HSEs are Cellular Component Organization or Biogenesis (GO:0071840) with 109 genes and Cellular Response to Stimulus (GO:0051716) with 95 genes (Appendix A). In these subcategories, the HSEs Ehpgp5 and Ehhsp100 are frequently found in the promoters of genes involved in cellular processes, component organization, and response to stimuli. The Metabolic Process category (GO:0008152), which includes 11 subcategories, is another major group with numerous genes containing HSEs. This category predominantly features genes encoding enzymes, such as the 26S Protease Regulatory Subunit (EHI_052050) with ATP hydrolysis activity, crucial for protein degradation in both cytosol and nucleus; Ribosomal RNA Methyltransferase (EHI_098730) with tRNA methyltransferase activity; Adenosine Deaminase (EHI_005040) with adenosine deaminase activity; and others like Methionine Gamma-Lyase (EHI_144610), Scavenger mRNA Decapping Enzyme (EHI_192480), and 2-Deoxy-D-Ribose 5-Phosphate Aldolase (EHI_121800), all involved in various metabolic processes in *E. histolytica*. The predominant HSEs in this category are also Ehpgp5, Ehhsp100, and Ehmlbp. These findings are consistent with those from the ontology of molecular functions, where catalytic activity had the highest number of genes with HSEs, and in biological processes, particularly in the Cellular Processes and Metabolic Processes categories.

We also observed genes with HSEs in other categories, although in smaller numbers. For example, the Biological Regulation category (GO:0065007) encompasses 142 genes, accounting for 15% of the total. Within this category, notable groups of genes include those encoding Myo-Inositol Monophosphatase (EHI_175410) and RING Zinc Finger Protein (EHI_023310). These genes are crucial for various processes, such as vesicular signaling, membrane trafficking, actin cytoskeleton formation, and cell adhesion. Predominantly, the promoters of these genes exhibit HSEs like Ehpgp5 and Ehhsp100 (Appendix A).

The fourth category within biological processes demonstrating a significant presence of genes is Response to Stimulus (GO:0050896), with 97 genes (10%). This group includes heat shock proteins (HSP) such as Heat Shock Protein (EHI_022620), HSP70 (EHI_188610), HSP90 (EHI_102270), and Peroxiredoxin (EHI_123390), which show oxidoreductase activity. These proteins are part of a widely distributed group of cysteine-dependent peroxidase enzymes, playing pivotal functions in maintaining peroxide levels in *E. histolytica*. Moreover, several genes in this category are involved in DNA repair, including DNA Repair Protein RAD51 Homolog (EHI_031220), RAD52 (EHI_188230), REV1 (EHI_053480), PMS1 (EHI_155170), and MLH1 (EHI_129950). Among the six key subcategories in this domain are Cellular Response to Stimulus (GO:0051716) with ninety-five genes, Response to Abiotic Stimulus (GO:0009628) with seven genes, Response to Chemical Stimulus (GO:0042221) with eight genes, Response to Endogenous Stimulus (GO:0009719) with two genes, Response to External Stimulus (GO:0009605) with two genes, and Response to Stress (GO:0006950) with forty-four genes. Remarkably, in the Response to Stress subcategory (GO:0006950), a substantial number of heat shock proteins are present, such as Heat Shock Protein (EHI_156560) and Heat Shock Protein (EHI_017350). These genes predominantly present the Ehpgp5 element, either as a single element or in combination with another HSE (e.g., Ehpgp5-Ehhsp100) (Appendix A).

### 2.8. Protein Classes

In the categorization of genes with HSEs within protein classes, they were divided into 15 categories. The top four categories, comprising the largest number of genes, include protein modifier enzyme (PC00260) with 171 genes, metabolite interconversion enzyme (PC00262) with 120 genes, protein binding modulator (PC00095) with 118 genes, and RNA metabolism protein (PC00031) with 74 genes (Figure 5c, Appendix A). Key genes in these categories are Guanine Nucleotide Exchange Factor (EHI_049610) and a GTP-Binding Protein (EHI_051090) in Protein Modifier Enzyme; Scavenger mRNA Decapping Enzyme (EHI_192480) and Adenosine Deaminase (EHI_005040) in Metabolite Interconversion Enzyme; TBC Domain-Containing Protein (EHI_166020) and Rap/Ran GTPase-Activating Protein (EHI_100290) in Protein Binding Modulator; and DNA-Directed RNA Polymerase Subunit (EHI_044620) and DEAD/DEAH Box Helicase (EHI_013960) in RNA Metabolism Proteins. In these categories, the elements Ehpgp5, Ehhsp100, and Ehmlpb were most frequently found. Moreover, genes involved in functions such as translation, membrane trafficking, DNA metabolism, and chaperoning were also identified (Appendix A). These findings indicate that genes with HSEs may play a significant role in the response of *E. histolytica* to various types of stress, impacting its molecular functions, biological processes, and cellular functions. The distribution of these genes across the categories, according to the six different HSEs, is illustrated in Appendix A.

In the first category of protein modifying enzyme (PC00260), four subcategories are identified: Non-Receptor Serine/Threonine Protein Kinase (PC00167) with 45 genes, Protease (PC00190) with 32 genes, Protein Phosphatase (PC00195) with 40 genes, and Ubiquitin-Protein Ligase (PC00234) with 18 genes. Representative genes from these subcategories are shown in Appendix A. Examples include Protein Kinase (EHI_186750) and Serine/Threonine-Protein Phosphatase (EHI_048590), primarily associated with phosphatase activity and calmodulin binding. In these genes, all six HSEs are found, with Ehpgp5 being the most prevalent.

The second major category is Metabolite Interconversion Enzyme (PC00262), comprising eleven subcategories involved in various activities. These include Hydrolase (PC00121) with 44 genes, Isomerase (PC00135) with 3 genes, Ligase (PC00142) with 6 genes, Lyase (PC00144) with 7 genes, Oxidoreductase (PC00176) with 21 genes, and Transferase (PC00220) with 39 genes. Within this category, genes like Acetyltransferase (EHI_096770), involved in O-acyltransferase activity, and Methionine Gamma-Lyase (EHI_144610), with functions in binding to heterocyclic or cyclic organic compounds and lyase activity, are found. Similar to the first category, Ehpgp5 is the most frequently observed HSE in these genes (Appendix A).

In the third category of protein-binding activity modulator (PC00095), five subcategories were identified: G-Protein Modulator (PC00022) with 59 genes, G-Protein (PC00020) with 47 genes, Kinase Modulator (PC00140) with 7 genes, Phosphatase Modulator (PC00184) with 1 gene, and Protease Inhibitor (PC00191) with 2 genes. Within this category, a prominent gene is the ARF GTPase-Activating Protein (EHI_069440), which is instrumental in vesicle targeting to, from, or within the Golgi apparatus, as well as in vesicle budding from membranes. This gene plays a key role in the transport, recognition, and binding of various proteins. Predominantly, the HSEs Ehpgp5 and Ehmlbp were detected in the promoters of these genes. These findings suggest that the ability of *E. histolytica* to respond to diverse stimuli may be attributed to the presence of HSEs in essential genes that are critical for its survival.

### 2.9. Expression Patterns of Genes with HSEs in Trophozoites under Different Stress Conditions

To determine whether genes in *E. histolytica* with HSEs in their promoter regions are actively expressed under different stress and culture conditions, we analyzed the 957 genes across various transcriptomes available on the Amoeba DB platform. These transcriptomes encompass trophozoites subjected to conditions such as starvation, oxidative stress, cyst-to-trophozoite conversion, and exposure to heat shock. Our analysis revealed that, in the transcriptome-specific cyst to trophozoite conversion [38], 262 out of the 1567 genes reported as upregulated contained HSEs. Specifically, Ehhsp100 emerged as the most prevalent element among these upregulated genes (Figure 6a). Detailed expression levels of these genes in this transcriptome are provided in Appendix A. It was observed that out of these genes, 220 have only one HSE (Table 4 and Table 5).

In this transcriptome, we identified genes such as Cys peroxiredoxin (EHI_084260) and ribosomal protein L13 (EHI_099730) that had an expression level 13 times higher than the control (Appendix A). Additionally, 31 genes featuring either two combined HSEs or the same HSE were identified. Prominent combinations include Ehhsp100-Ehmlbp and Ehhsp100-Ehpgp5, found in the promoters of Protein Kinase (EHI_035500) and Rap/Ran GTPase-Activating Protein (EHI_100290), respectively, both showing over nine-fold overexpression. Nine genes exhibited combinations of three HSEs. For instance, the *enolase* gene (EHI_130700) with Ehhsp100-Ehpgp5-Ehmlbp increased its expression 12 times compared to unstressed levels. Similarly, the *actin* gene (EHI_182900) with Ehhsp100-EhrabB1-Ehmlbp and *hsp90* (EHI_196940) with one Ehhsp100 and two Ehmlbp elements demonstrated 10.3 and 10.8 times higher expression than the control, respectively. Remarkably, two genes had four HSEs. The Vesicle-Associated Membrane Protein (EHI_081370), involved in synaptic vesicle coupling and fusion, exhibited four Ehhsp100 elements and showed a 10-fold increase in expression. The beta subunit of Phenylallyl-tRNA Synthetase (EHI_002130) presented with three Ehpgp5 and one Ehhsp100 elements, also achieving a 10-fold elevation in expression compared to reference levels (Appendix A). This analysis highlights that, under stress conditions, several genes vital for trophozoite survival may be regulated by specific HSEs. Ehhsp100 and Ehpgp5 emerge as key regulatory elements in this context, potentially playing a crucial role in the adaptive response of trophozoites to stress.

In the context of intestinal invasion, where a reported upregulation of 1566 genes occur [39], we identified 269 genes containing HSEs. Figure 6a, Appendix A details the expression levels of these genes in this transcriptome. Among them, 225 genes possess a single HSE element, with Ehhsp100 being again the most abundant HSE in the overexpressed genes, present in 102 genes (Table 4 and Table 5). Prominent examples include the Ribosomal S18e gene (EHI_031400), which is believed to act as a molecular clamp binding to 18S rRNA and shows a 10-fold increase in expression. Similarly, the gene for the Gal/GalNAc lectin subunit Igl2 (EHI_065330) with an Ehpgp5 element exhibits a 12.26-fold increase (Appendix A).

Furthermore, 34 genes have two unique HSEs or in combination. In this case, the gene encoding the pore-forming peptide amebapore A precursor (EHI_159480) with two EhrabB1 elements shows a 12.7-fold increase in expression. The gene for *hsp90* protein (EHI_102270), vital in stress response, containing Ehhsp100-Ehmlbp elements, is expressed 10.7 times more. Other critical genes identified include cysteinyl-tRNA synthetase (EHI_169700) with two Ehhsp100 elements, showing a 10.1-fold increase, and a nucleoside transporter (EHI_110730) with two Ehpgp5 elements, expressed 8.8 times higher. Furthermore, eight overexpressed genes in the intestinal invasion condition possess three unique or combined HSEs. Interestingly, three of these, including *enolase* (EHI_130700), *actin* (EHI_182900), and *Histone H3* (EHI_193360), are also overexpressed in the nutrient-starvation condition, with expression increases of 11.5, 12.3, and 8.9 times, respectively (Appendix A).

A similar pattern is observed with only two genes that have four HSEs and are overexpressed during intestinal invasion. These include the Vesicle-Associated Membrane Protein function (EHI_081370), which is overexpressed 9.7 times, and the Phenylalanyl-tRNA Synthetase Beta Subunit (EHI_002130), showing a 10-fold increase in expression. These results imply that these genes are essential in responding to the conditions of cyst-to-trophozoite conversion and intestinal invasion. They play vital roles in cellular translation and as post-transcriptional regulators, involved in cold sensitivity, phosphate transport, and virulence, among other functions [40,41,42]. Furthermore, in the contexts of both cyst-to-trophozoite conversion and starvation, there are 62 genes upregulated in both conditions with HSEs in their promoters. An example is the 60S Ribosomal Protein L37 (EHI_158270) with Ehhsp100, which exhibited expression increases of 12.7 and 11.37 times in the conversion and starvation transcriptomes, respectively (Figure 6b).

Under nutrient-starvation stress, a significant change in gene expression was noted, with 1647 genes reported to be affected [43]. Within this group, 241 genes were identified as having HSEs in their promoters (Figure 6a, Table 4). Among these, 114 genes are downregulated, including 94 with a single HSE. Notable examples of such genes are the Actinin-like Protein (EHI_155290) with Ehhsp100, and the Transcription Initiation Factor IIIB Chain BRF (EHI_158020) with Ehmlbp (Appendix A). Additionally, 15 genes have either two unique or combined HSEs. For example, Nucleosome Assembly Protein 1-Like 1 (EHI_006530) contains two Ehhsp100 elements, and Serine-Threonine-Isoleucine Rich Protein (EHI_025700) features Ehhsp100-EhrabB4, with Ehhsp100 being the most frequent element in this group (Table 5). Moreover, only five genes exhibit three HSEs. These include Formin Homology 2 Family Protein (EHI_192460) and hsp90 (EHI_196940) with one Ehhsp100 and two Ehmlbp, WD Domain Containing Protein (EHI_052160) with three Ehhsp100, *enolase* (EHI_130700) with Ehpgp5-Ehhsp100-Ehmlbp, and *actin* (EHI_182900) with Ehhsp100-EhrabB1-Ehmlbp. Intriguingly, in this nutrient-starvation condition, the last two genes, *enolase* and *actin*, exhibit downregulation, contrasting their expression patterns in the conversion and intestinal invasion transcriptomes.

Moreover, under nutrient-starvation stress, in *E. histolytica*, there must also be a highly coordinated response for the survival of this parasite. It has been observed that trophozoites exposed to this stress increase their virulence and modify their gene expression [43,44]. Interestingly, in this study, we found that overexpressed genes, such as the one encoding the dihydropyrimidine dehydrogenase enzyme crucial in pyrimidine metabolism, as well as the surface antigen Ariel 1, the galactose-inhibitable lectin small subunit, some 60S ribosomal subunits—all crucial for the parasite’s growth under this stress situation—have one to four Heat Shock Elements (HSEs). Therefore, their expression could be directly correlated with the presence of these HSE elements. The same may occur with genes whose expression is downregulated, such as Enolase (EHI_130700), which has been observed to play an important role in glycolysis [45], and several ribosomal proteins that are involved in growth regulation, cell proliferation, and DNA damage response [46,47,48].

We investigated which genes contained HSEs in the transcriptome of trophozoites exposed to 2, 4, and 8 h heat shock at 42 °C. The study reported a total of 2210 genes exhibiting varied expression profiles, categorized into four groups: no expression, low expression, medium expression, and high expression [49]. In analyzing the 957 genes with HSEs for the 2 h expression profile, we found that 256 genes displayed no change in expression levels. In this group, the Ehhsp100 element was most frequently observed, with 89 occurrences (Table 4 and Table 5). However, 587 genes showed low expression levels (1 to 99-fold), with the Ehpgp5 element (n = 193) being the most common in this category (Table 5). Within this group, 486 genes had a single HSE, including Ribosome Biogenesis Protein (EHI_175090) and Small GTPase Rab11B (EHI_107250). Additionally, 80 genes featured two HSEs, with the Ehpgp5-Ehhsp100 combination being particularly prevalent. This combination was found in genes such as Plasma Membrane Calcium-Transporting ATPase (EHI_030830) and Vesicle-Associated Membrane Protein (EHI_170630). A total of 17 genes with three HSEs were identified, including Heat Shock Protein 90 (EHI_196940) and RuvB-like DNA Helicase (EHI_091070). Two genes had four HSEs: Phenylalanyl-tRNA Synthetase Beta Subunit (EHI_002130) and IBR Domain Containing Protein (EHI_120320). Additionally, two genes contained five HSEs: DEAD/DEAH Box Helicase (EHI_169630) and Cell Division Cycle Protein 123 (EHI_128440) (Appendix A). Furthermore, 78 genes exhibited moderate expression levels (100- to 1000-fold increase), with Ehhsp100 (n = 35) being the most prevalent HSE in this category. Among these, 63 genes had a single HSE, including Glucosamine-6-Phosphate Isomerase (EHI_174640) and Adenylate Kinase (EHI_135470). Twelve genes featured two HSEs, with the EhrabB4-Ehhsp100 combination occurring most frequently, as seen in genes like *actin* (EHI_182900) and Enhancer Binding Protein-2 (EHI_182670). Two genes had three HSEs, such as the WD Domain Containing Protein (EHI_052160), while only the Vesicle-Associated Membrane Protein gene (EHI_081370) had four HSEs. Finally, 36 genes displayed high expression levels (over 1000-fold increase), with Ehhsp100 (n = 18) being the most common HSE in this category (Table 4 and Table 5). Among these high-expression genes, 30 had a single HSE, such as Ubiquitin-Like Protein (EHI_170060) and 60S Ribosomal Protein L24 (EHI_030760) (Appendix A).

Five genes were identified with two HSEs, including Ribosomal Protein S18 (EHI_009680) and 60S Ribosomal Protein L19 (EHI_110340). Additionally, one gene, *enolase* (EHI_130700), was found to have three HSEs (Figure 7a). Interestingly, a pattern emerges from these data: genes with medium and high expression levels predominantly contain the Ehhsp100 element in their promoters, while those with low expression levels frequently have the Ehpgp5 element. This trend suggests a potential correlation between the type of HSE present and the gene’s expression level.

At 4 h of exposure to heat shock, the distribution of genes with HSEs across the four expression groups shifted, with an increase in the low expression group and decreases in the no expression, medium expression, and high expression groups. In this scenario, 221 genes remained unchanged in their expression, presenting both Ehhsp100 (n = 75) and Ehpgp5 (n = 75) as the most prevalent elements in equal proportion within this group (Table 4 and Table 5). Meanwhile, 630 genes were categorized in the low expression group, with the Ehpgp5 element (n = 205) being the most frequent. Among these, 517 genes possessed a single HSE, including Lipid Phosphate Phosphatase (EHI_024410) and WD Domain-Containing Protein (EHI_192950). Additionally, 90 genes showed two HSEs, such as *Helicase* (EHI_141120) and RhoGAP Domain-Containing Protein (EHI_036880), predominantly with the Ehpgp5-Ehhsp100 combination. A total of 19 genes had three HSEs, including AAA Family ATPase (EHI_004190) and Tyrosine Kinase (EHI_065500). Additionally, two genes displayed four HSEs, and another two genes contained five HSEs (Table 4, Appendix A).

In the medium expression level group, 74 genes were identified, most with only one HSE (n = 61). The Ehhsp100 element was the most common in this group (n = 33) (Table 5). Ten genes had two HSEs, with Ehhsp100-EhrabB4 being the prevalent combination. Two genes contained three HSEs, such as *actin* (EHI_182900), and only one gene had four HSEs.

Furthermore, 32 genes demonstrated high expression levels. Among these, 26 had a single HSE, with Ehhsp100 (n = 15) being the most frequent. Five genes had two HSEs, often presenting the Ehmlbp element in combination with one of the other five HSEs. These included Actin-Binding Protein (EHI_186840) with Ehmlbp-Ehpgp5 and 60S Ribosomal Protein L9 (EHI_193080) with Ehmlbp-EhrabB1. Remarkably, only one gene, *enolase* (EHI_130700) with three HSEs (Ehpgp5-Ehhsp100-Ehmlbp), exhibited a 13191.29-fold increase in expression. This gene also showed significantly high expression levels (12130-fold and 11652.2-fold increases) at both 2 and 4 h of heat shock exposure (Figure 7b). However, it is important to note that the gene with one of the highest expression levels (12790.17-fold) is the precursor of the pore-forming peptide amoebaporin A (EHI_159480), which plays a crucial role in pore formation in the host cell membrane and contains two EhrabB1 elements.

Finally, upon 8 h of heat shock exposure, there was a notable shift in gene expression profiles, with an increase in the number of genes in the low, medium, and high expression categories (Table 4), albeit with a decrease in overall expression levels (Appendix A). Among them, 168 genes exhibited no change in mRNA expression, with Ehhsp100 (n = 60) being the predominant HSE in this group (Table 5). In the low expression group, out of 653 genes, Ehpgp5 (n = 213) was the most common element. This group included 538 genes with a single HSE, such as Leucine-Rich Repeat Protein, BspA Family (EHI_070230), and Protein Kinase (EHI_197540). Ninety-three genes had two HSEs, including 26S Protease Regulatory Subunit (EHI_052050) and Rab GTPase Activating Protein (EHI_009970). Additionally, eighteen genes had three HSEs, two genes had four, and two had five HSEs.

In the medium expression level group, 103 genes were identified, with Ehhsp100 (n = 46) being the most frequent HSE. Within this category, 90 genes had a single HSE, such as 60S Ribosomal Protein L13 (EHI_181560) or RNA Recognition Motif Domain-Containing Protein (EHI_026440). Ten genes showed two HSEs, like G Protein Alpha Subunit (EHI_140350) or RNA-Binding Protein (EHI_151990). Additionally, two genes contained three HSEs, and one gene had four.

For the high expression group, 33 genes were noted, with Ehhsp100 (n = 14) consistently being the most common HSE, keeping consistent with the trends observed at the 2 and 4 h expression levels. This group included 26 genes with a single HSE, such as 40S Ribosomal Protein S26 (EHI_105180) and Pyruvate:Ferredoxin Oxidoreductase (EHI_051060). Six genes had two HSEs, and two genes presented three HSEs (Figure 7c, Table 4 and Table 5, Appendix A).

We also investigated whether there was a correlation between the expression levels and the positioning of HSEs in genes exhibiting higher expression at the three time points of 2 h, 4 h, and 8 h (Figure 7d).

Our results revealed that there was no specific pattern in the positioning of these elements; they were distributed throughout the promoter region (−1 to −500 bp). The findings regarding the presence of HSEs in the promoters of genes in trophozoites, which alter their expression under thermal stress, indicate that the expression levels of these genes do not seem to be dependent on the number of HSEs. Rather, they may be influenced by the specific type of HSE present. Relevantly, the most common elements in genes that were highly expressed were Ehhsp100 and Ehmlbp (Table 5).

Moreover, the expression levels of upregulated genes varied significantly across the three different time points of heat shock. Notably, the expression level at the 8 h (68,000) was not as elevated compared to those observed at the 2 and 4 h time points (131,000). This variation suggests that prolonged stress exposure may lead to a decrease in overexpression levels, or that the expression of these genes reaches a state of stabilization over time. Among these genes, *enolase* (EHI_130700) consistently remained upregulated across all time points. Given its significant role in the differentiation of *E. histolytica* [50], this consistent upregulation provides a plausible explanation for its observed expression pattern.

### 2.10. 3D Modeling and Structural Validation of tEhDBD5, tEhDBD6, and tEhDBD7 Homotrimers

To determine whether *E. histolytica* heat shock transcription factors (EhHSTFs) specifically bind to the six different HSEs identified (Ehpgp5, Ehhsp100, Ehmlbp, EhrabB1, EhrabB4, and EhrabB5), *blind* molecular dockings were performed in silico. Initially, models of the transcription factors EhHSTF5, EhHSTF6, and EhHSTF7 were constructed. Particularly, the DNA binding domains (DBDs) of these transcription factors in their homotrimer conformation were modeled based on the crystallized structure of human HSF1 complexed with a DNA sequence containing three HSE motifs. Comparative analysis revealed that human HSF1 shares a significant amino acid identity with the EhDBDs: 46.88% with EhDBD5, 40.43% with EhDBD6, and 50% with EhDBD7, as indicated by the results from SWISS-MODEL. The 3D models of the transcription factors were rigorously validated using various complementary in silico tools. Initially, Ramachandran analysis on the PDBsum server was performed to identify potential structural anomalies based on the dihedral angles of the peptide bonds. The analysis showed that 100% of the amino acids in the models reside in allowed regions, with G average factors of −0.09, −0.10, and −0.04 for tEhDBD5, tEhDBD6, and tEhDBD7, respectively. These values, being close to zero, signify excellent structural integrity.

Furthermore, the Prosa-Web server analysis yielded energy values of −6.11 for tEhDBD5, −3.58 for tEhDBD6, and −6.2 for tEhDBD7. These scores are consistent with the ranges typically observed in structures determined using X-ray crystallography and nuclear magnetic resonance (NMR), indicating reliable model quality. The overall energy analysis revealed that all amino acids possess negative energy values, suggesting the absence of structural errors related to global energy.

In addition, the ERRAT tool was used to identify structural errors based on non-bonded atom–atom interactions, resulting in Overall Quality Factor scores of 86, 89.95, and 88.09 for tEhDBD5, tEhDBD6, and tEhDBD7, respectively. VERIFY 3D further evaluated the physicochemical properties of the amino acids in the models, indicating that 84.59% of amino acids in tEhDBD5, 96.38% in tEhDBD6, and 84.72% in tEhDBD7 have a 3D-1D average score ≥ 0.1. This implies that more than 80% of the residues in these DBDs are well conformed and align with characteristics typical of crystallized protein structures (Figure 8, Appendix A).

Simultaneously, 3D models for each of the six HSEs of *E. histolytica* were obtained using the UCSF Chimera software (version 1.17). These models were constructed based on the following nucleotide sequences: 5′-ataGAAatttTTCata-3′ for the Ehpgp5 gene’s HSE, 5′-aagGAActtGAAgaa-3′ for Ehhsp100, 5′-agaaGAActaataactGAAaattataatTTC-3′ for Ehmlbp, 5′-cTTCaTTCcatggtttTTC-3′ for EhrabB1, 5′-gTTCttcagtaagaGAA-3′ for EhrabB4, and 5′-aTTCtacaaaaaGAA-3′ for EhrabB5.

### 2.11. Molecular Docking and Intermolecular Analysis

Considering that HSFs are known to bind to DNA as trimers, we performed molecular docking analyses using the homotrimer conformation of the EhDBDs contained in the EhHSTF5, EhHSTF6, and EhHSTF7 factors. For visual aid, we color coded these monomers: monomer 1 in red, monomer 2 in green, and monomer 3 in yellow, as illustrated in Figure 9 and Appendix A. The docking results consistently indicated that tEhDBD7 has the highest binding affinity to all six HSEs tested, in terms of bonding affinity, followed by tEhDBD5, and then tEhDBD6 (Figure 9a). Owing to tEhDBD7’s stronger binding preference and its greater similarity to the DBD of human HSF1, we concentrated our analysis on elucidating the binding mechanism between tEhDBD7 and the various HSEs (Figure 9b–g).

The molecular docking analyses demonstrated a significant binding affinity of tEhDBD7 towards various HSEs. The binding affinities of tEhDBD7 to various HSEs, quantified by HDOCK, showed docking scores of −502 for the EhrabB1 HSE promoter, −497 for Ehhsp100, −489 for Ehmlbp, −488 for EhrabB4, −474 for Ehpgp5, and −446 for EhrabB5 (Figure 9a). A notable observation was the insertion of alpha helix 3 (α3-helix) of tEhDBD7 into the major groove of the HSEs, indicating its pivotal role in facilitating key interactions.

Intermolecular analysis shown in Appendix A revealed that tEhDBD7 binds to the HSE of the Ehpgp5 promoter through 42 interactions, including 6 hydrophobic, 28 hydrogen bonds, and 8 salt bridges. In the case of the Ehhsp100 HSE, 35 interactions were identified, composed of 3 hydrophobic, 27 hydrogen bonds, 2 salt bridges, and 3 π–Cation interactions. For the Ehmlbp HSE, 48 interactions were observed, including 5 hydrophobic, 31 through hydrogen bonds, 10 salt bridges, and 2π–Cation interactions. In the EhrabB1, 39 interactions were identified, broken down into 3 hydrophobic, 25 hydrogen bonds, 9 salt bridges, and 2π–Cation interactions. Likewise, the EhrabB4 HSE showed 41 interactions, with 3 hydrophobic, 25 hydrogen bonds, 10 salt bridges, and 3 π–Cation interactions. Finally, the EhrabB5 HSE presented 39 interactions, consisting of 4 hydrophobic, 24 hydrogen bonds, 9 salt bridges, and 2π–Cation interactions.

In our intermolecular analysis, we focused specifically on the GAA and TTC motifs within the HSEs of various promoters (Figure 9h).

We observed that the tEhDBD7 domain shows a remarkable affinity for the 3′-5′ sequence of the HSE of the Ehpgp5 promoter. In this context, monomer 1 interacts with the amino acids Thr16, Phe17, His64, Asn66, Ser69, Gln73, Tyr77, Ile76, and Arg108. Conversely, monomer 3 establishes contacts with Lys63, Lys81, Asn75, Arg72, and Gln88. Particularly, monomer 2 does not participate in these interactions, and no significant interactions were detected for the motifs in the 5′-3′ sequence.

For the HSE of the Ehhsp100 promoter, monomer 1 binds to Phe17, Lys63, His64, Asn66, Ser69, Gln73, Lys81, and Arg108 in the 5′-3′ sequence. Monomer 2 interacts with His80 and Lys81, while monomer 3 associates with Lys63. In the 3′-5′ sequence, monomer 1 interacts with Lys63, Arg72, and Asn75, and monomer 2 with Phe17, Lys63, His64, Asn66, Ser69, Gln73, and Arg108. However, no interactions were observed for monomer 2 with the motifs in this sequence.

Regarding the Ehmlbp promoter’s HSE, monomer 1 does not directly interact with the GAA and TTC motifs in the 5′-3′ sequence. However, monomer 2 binds to Arg72, Asn75, His80, and Lys81, and monomer 3 to Lys63 and Lys81. In the 3′-5′ sequence, only monomer 2 recognizes the GAA motif, interacting through Arg71 and Asn75.

In the case of the EhrabB1 promoter’s HSE, the amino acids Arg72, Asn75, His80, and Lys81 of monomer 2 interact with the 5′-3′ sequence. Monomer 3 contributes with Lys81 and Gln88, while in the 3′-5′ sequence, monomer 1 binds through Arg74 and Asn75, with monomer 3 interacting only with His84.

In relation to the EhrabB4 promoter, the GAA and TTC motifs in the 5′-3′ sequence are recognized by Thr16, Phe17, Lys63, His64, Gln73, Tyr77, and Arg102 of monomer 2. This monomer also binds to Arg72, Asn75, and Lys81, while monomer 3 interacts with Arg72, Asn75, Lys81, Arg102, and Gln88. In the 3′-5′ sequence, Arg72 and Asn75 of monomer 2 and Thr16, Phe17, Gln73 of monomer 3 are involved, though monomer 2 shows no interaction in this sequence.

Finally, for the EhrabB5 promoter’s HSE, in the 5′-3′ sequence, only Arg72, Asn75, and Lys81 of monomer 2, along with Asn75, Lys81, and Gln88 of monomer 3, are involved in the bind. In the 3′-5′ sequence, only Arg72 and Asn75 of monomer 1 interact.

These results suggest that the six HSEs identified in the promoter regions of 957 *E. histolytica* genes can potentially be recognized by three of the seven EhHSTFs present in this parasite and regulate expression efficiently under various conditions faced by the parasite, ensuring its survival. This mechanism points to a sophisticated gene regulatory system within *E. histolytica*, highlighting the adaptive capacity of this parasite at the molecular level.

## 3. Discussion

The stress response is crucial for cellular survival under adverse conditions. All organisms, including parasites like *E. histolytica*, encounter various types of stress throughout their lifecycle. This amoeba transitions between the cyst and trophozoite phases, exposing itself to multiple challenges within its host. These challenges include digestive enzymes, the immune system, variations in cell types, pH, and temperature, among others [51]. The amoeba must respond rapidly to survive these conditions. Each response involves transcriptional changes that enable the amoeba to repress or express specific genes, aiding its survival under particular circumstances [52]. To coordinate these responses, regulatory elements are assumed to exist in the promoters of certain genes. These elements allow the organism to react to specific stimuli, along with master transcription factors that recognize and ensure their expression [53]. An example is the thermal stress response, where HSTFs are activated in response to temperature increases. These HSTFs recognize HSEs in the promoters of specific genes, such as HSP chaperones, inducing their expression to prevent protein denaturation [54].

The objective of the research included screening, detection, and identification of HSEs in the promoters of the *E. histolytica* genome, as well as the structural characterization of these elements, and the establishment of a correlation between the presence of HSEs and the gene expression in trophozoites cultured under different stress conditions, as well as predicting DNA–protein interactions between factors EhHSTF5, EhHSTF6, and EhHSTF7 with the six different HSEs.

The amoeba, identified as a protozoan parasite, encompasses a total of 8343 genes. Notably, functional characterization has been conducted on the promoters of less than 1% of these genes [28,31,55]. In our research, we discovered a total of 2578 HSEs located in the promoter regions of 1967 coding genes within the *E. histolytica* genome. This indicates that at least 24% of the genes in this parasite’s genome contain HSEs. Such a significant proportion suggests that *E. histolytica* may have the capability to regulate the expression of over a quarter of its genes in response to various stimuli, such as heat stress, nutrient deprivation, immune challenges, and exposure to drugs. Relevantly, in several studies, HSEs have been identified in gene promoters in different organisms, including *H. sapiens* [56,57], *S. cerevisiae*, *D. melanogaster* [17], *M. musculus* [58], *Cryptosporidium parvum* (*C. parvum*) [59], and *C. elegans* [60], and in various plants such as *A. thaliana* [22], *Triticum aestivum* L. (*T. aestivum*) (wheat) [61], *Zea mays* (*Z. mays*) (corn) [62], and *Solanum lycopersicum* (*S. lycopersicum*) (tomato) [63]. This research underscores the importance of HSEs in facilitating the regulation of diverse genes across different conditions, a key factor in processes vital to organisms.

The exploration of these sequences, however, has been limited in scope, focusing primarily on the promoters of genes responsive to thermal stress, rather than encompassing the entire genome. This approach is exemplified in the research by Cohn et al. [59] on *C. parvum*, where they discovered HSEs in 11 out of 12 gene promoters associated with heat shock response, including HSPs. Similarly, Arce et al. [64] identified two specific HSEs, HSF1AE and HSF21, in the promoter regions of a group of nine HSPs in tomatoes.

In a more recent study, Zhao et al. [65] performed a pioneering genome-wide analysis of *T. aestivum* L., discovering 39,478 HSEs within 30,604 genes. This represents 27.6% of the total genome, a similar proportion to that observed in our study. Notably, they found that HSEs were not limited to the promoters of genes responding to thermal stress, like HSPs. Instead, HSEs were also present in genes related to other responses, such as biotic stress, oxidative stress, and various biological processes including differentiation, development, and metabolism.

Consistent with these findings, our study also identified HSEs predominantly in the promoters of various enzymes, transporters, and assembly proteins. These proteins play crucial roles in essential processes such as replication, transcription, translation, virulence, migration, and metabolism. Furthermore, our results corroborate the observation that HSEs extend beyond the promoters of genes responding to heat shock. This is consistent with HSE elements found in the promoter regions of genes involved in immune responses, as has been documented in certain interleukins [32], drug resistance genes such as *mdr* [28], and enzymes implicated in oxidative stress, including Heme oxygenase-1 HMOX1 [20,66]. This demonstrates the varied and wide functional roles of HSEs in genomic regulation.

Therefore, these results provide a new perspective on the transcriptional regulation of organisms through HSEs, both in response to stress and in different processes essential for survival. However, to deepen our understanding of the importance of these responsive elements in stimulus response and general transcription processes, additional research is needed to identify these elements in the genomes of more organisms. This study marks the second exhaustive analysis of this nature.

Moreover, the number of HSEs in the promoters of *E. histolytica* revealed that the majority exhibit a single HSE, as reported in most genes across various organisms, such as the *H. sapiens mdr1* gene (−178 to −152 bp), the *D. melanogaster hsp70* gene (−220 to −150 bp), or the *apx2* gene (−282 to −346) in *A. thaliana* [67]. Nevertheless, and albeit in smaller numbers, genes with two different HSEs have also been identified, such as *cup1* (−172 to −143 bp and −134 to −120 bp), a metallothionein in *S. cerevisiae*, or an *hsp70* (−450 to −437 bp and −218 to −203 bp) in *C. elegans* [23,68]. In *E. histolytica*, several genes exhibited two HSEs. These include the amebapore gene (EHI_159480), a crucial virulence factor that induces pore formation, and the *helicase* gene (EHI_141120), pivotal in replication and transcription processes. Additionally, genes encoding basal transcription factors like TFIID (EHI_137090) and TFIIH (EHI_182880), as well as various HSPs (EHI_193390, EHI_070450, EHI_102270, EHI_013550), were found to have two HSEs. The presence of multiple HSEs in the genes is not unique to *E. histolytica*. Comparable cases have been described in other organisms. In *S. cerevisiae*, for example, the gene responsible for encoding HSP70 contains three HSEs, as detailed by Young et al. [69]. Similarly, in *H. sapiens*, the gene encoding IL-8, identified by Smith et al. [70], has been found to possess two HSEs. Mutations and deletions in HSEs have shown that the removal of a single motif significantly decreases the expression of both the *hsp70* and *IL-8* genes, highlighting the critical importance of these elements.

Additionally, in *E. histolytica*, a groundbreaking discovery was made, revealing for the first time that two promoters contain up to five responsive elements. These genes code for a DEAD/DEAH box helicase, putative (EHI_169630). This protein plays a key role in catalyzing the folding and remodeling of RNA molecules across prokaryotic and eukaryotic cells [71] and the cell division cycle protein 123 homolog (EHI_128440), which regulates the cell cycle in a nutrient-dependent manner [72]. This finding implies that such genes are crucial and require consistent transcription under various conditions. The presence of five HSEs in these genes could be a mechanism to ensure their transcription across different environments, or it might indicate that at certain times, high levels of their transcripts are necessary, thereby necessitating multiple HSEs to enhance their expression.

Interestingly, studies like the one conducted by Zhao et al. [65] have observed that variations in the magnitude of the expression response of some genes to heat stress correlate directly with the structure of HSEs and, importantly, their quantity. In certain genes, such as *cup1* in *S. cerevisiae* or *hsp70* in *C. elegans*, which have two or three HSEs, there is a demonstrated direct relationship between the number of HSEs and the quantity of transcript produced.

The quantity of HSEs plays a pivotal role in gene recognition and subsequent transcription. This observation is consistent with findings by Naiyer et al. [73], which highlighted that the gene encoding for amebapore exhibits the highest expression levels in both the virulent strain isolated from a patient and in the *E. histolytica* strain passed through the liver. This elevated expression could be linked to the two HSEs present in its promoter. However, to fully understand the impact of these elements, it is crucial to conduct a functional analysis of their role in the promoter, especially under varying stress conditions.

In *E. histolytica*, our analysis revealed that the GAA and TTC motifs within HSEs are highly conserved. We identified HSEs containing either two (1745 instances) or three motifs (834 instances), with a higher prevalence of HSEs comprising three motifs. This finding is consistent with the identification of HSEs in other organisms, where the GAA and TTC motifs are critical for the recognition of HSFs. This pattern is not unique to *E. histolytica* but is also observed in various other species, including *D. melanogaster*, *S. cerevisiae*, *E. coli*, *Stratiomys singularia* (*S. singularior*), and *H. sapiens* [27,74,75,76].

In a recent study, we demonstrated the interaction between the EhHSTF7 factor and the Ehpgp5 element in the promoter of the multidrug resistance (*mdr*) gene *Ehpgp5* in *E. histolytica*. This interaction is particularly focused on the 5′-GAA-3′ motif found on the complementary strand. Our findings revealed that point mutations within this motif impede the recognition of EhHSTF7 by the HSE. Moreover, silencing EhHSTF7 resulted in a marked reduction in both the gene and protein expression levels of *Ehpgp5*. This indicates that the interaction between EhHSTF7 and the HSE is crucial for activating the expression of the *mdr* gene *Ehpgp5* in trophozoites when they are exposed to the drug emetine [27].

Likewise, in 2016, Nieto et al. [28] demonstrated the indispensability of the HSE in inducing the expression of the MDR gene *Ehpgp5* in *E. histolytica* trophozoites exposed to the anti-amoebic drug emetine by deleting and mutating the HSE in the promoter of the *Ehpgp5* gene. These results highlight the essential role of the structural organization and functionality of HSEs within the promoter of an inducible gene such as Ehpgp5, especially under stress conditions like drug exposure. Moreover, the studies accentuate the crucial function of EhHSTF7 in *E. histolytica*, underscoring its substantial impact on gene expression in response to a range of stressors.

The characteristics of HSEs, including their size, motif arrangement, and the variation in the number of nucleotides between motifs, are consistent with descriptions of HSEs previously identified in various organisms like *H. sapiens*, *D. melanogaster*, *S. cerevisiae*, and *C. elegans* [24,60,69]. Additionally, the creation of a Position Weight Matrix (PWM) for each HSE has robustly supported these findings by enabling the identification of nucleotide preferences at each position within each element. This probabilistic model has been instrumental in identifying binding sites for various transcription factors, such as p53, the estrogen receptor, and TLX3, among others, as demonstrated by Pujato et al. [77].

The location of transcription factor binding sequences is an important aspect to consider. For HSEs, a specific location is not universally defined. However, studies on *D. melanogaster*, particularly regarding the promoter of the *hsp70* gene, have shown that altering the position of HSEs can lead to reduced promoter activity [24]. In *E. histolytica*, the identified HSEs in gene promoters are precisely located between positions −1 to −624. The HSE in the promoter of the *Ehpgp5* gene, specifically situated at −150 to −137 bp, has been the subject of functional studies [28]. Furthermore, the deletion of HSEs in EhRabB1, EhRabB4, and EhRabB5 results in decreased expression of the reporter gene in trophozoites subjected to thermal shock [31].

Upon searching for previously described regulatory sequences TATA, INR, and GAAC in *E. histolytica*, it was observed that only 36% of the genes with HSEs contain one of these three sequences. Notably, the GAAC sequence, unique to *E. histolytica*, was the most prevalent, suggesting its potential significance in transcription within this parasite. These results agree with those published by Naiyer et al. [73]. Although the presence of any of these regulatory elements could enhance the precision of transcription initiation and potentially enhance the expression of some genes, there is currently no study in amoeba that confirms this. Therefore, it is necessary to investigate whether deleting or mutating any of these sequences would alter transcription initiation or gene expression.

It is evident that not all genes with HSEs are activated or repressed simultaneously. This was confirmed by analyzing the expression of the 957 genes we identified with HSEs in the reported transcriptomes of *E. histolytica* trophozoites exposed to different stress conditions. Interestingly, we observed that 16.7%, 17.1%, and 15.3% of the genes that altered their expression in the transcriptomes during cyst-to-trophozoite conversion, intestinal invasion, and nutrient starvation, respectively, contain HSEs. Meanwhile, in trophozoites exposed to thermal stress for 2, 4, and 8 h, the percentage of genes with HSEs that changed their expression was double, with respective percentages of 31.7%, 33.3%, and 35.7%.

First, these results suggest that a considerable number of genes containing HSEs play a crucial role in responding to a variety of stimuli, with their expressions and levels potentially being regulated by these elements. Second, it is noteworthy that the genes with HSEs, which show altered expression in these transcriptomes, are not exclusively associated with thermal stress responses. In fact, 82 genes were consistently identified across all four examined transcriptomes, encompassing thermal shock, intestinal invasion, cyst-to-trophozoite conversion, and nutrient starvation. Notable among these genes are Fe-hydrogenase (EHI_073390), 40S ribosomal protein S10 (EHI_197030), and Rho guanine nucleotide exchange factor (EHI_005910), indicating a broader functional spectrum of genes influenced by HSEs, beyond the stress response.

Fascinatingly, these genes exhibit one to four HSEs, with the majority harboring a single HSE, predominantly Ehhsp100. This suggests their activation is likely mediated through the HSE, possibly by the same EhHSTF7. These findings indicate that a gene can be regulated by multiple HSEs, as reported in other organisms [24]. Third, genes with HSEs show significantly higher expression levels compared to those without HSEs. However, the expression level among genes with one or more HSEs (up to five) does not demonstrate a directly proportional relationship. For instance, in the cyst-to-trophozoite conversion transcriptome, genes with HSEs showed an 8- to 12fold increase in expression level compared to genes lacking HSEs, despite having one to four HSEs.

The results of this research demonstrate that HSEs regulate not only genes involved in the heat shock response, such as HSPs, but also genes encoding a spectrum of proteins, including helicases, kinases, phosphatases, hydrolases, elongation factors, transporters, and cytoskeletal proteins, which may be integral to the stress response mechanism. This finding agrees with the screening of HSEs in the promoter regions of genes implicated in functions previously considered distinct. Prominent examples include regulation of drug resistance and transport as observed in *Ehpgp5* [78], heat stress response in Ehhsp100 [29], methyl-binding protein in Ehmlbp [30], and intracellular vesicle transport in EhrabB [31].

In *H. sapiens*, it has been shown that genes such as *cox-2*, which encodes cyclooxygenase-2 (COX-2), are involved in regulating the inflammatory response [79]. Additionally, the type I multidrug resistance gene, MDR1, is regulated by a HSE [80]. These findings suggest that HSEs not only regulate genes associated with stress response, but they also play a vital role in ensuring the survival of the organism.

Interestingly, the results of molecular docking analysis revealed that the factors EhHSTF5, EhHSTF6, and EhHSTF7, acting as homotrimers, are able to recognize the HSEs of the Ehpgp5, Ehhsp100, Ehmlbp, EhrabB1, EhrabB4, and EhrabB5 genes of *E. histolytica*, identified as of this moment. This suggests that, under different conditions, the parasite could ensure the expression not only of these genes but also of other genes that are regulated by an HSE, enabling it to respond efficiently to different conditions.

Recently, a study by Smith et al. [70] revealed similar results to ours, specifically in relation to HSF1 and HSF2 factors in *H. sapiens*. Furthermore, in 2009, research by Yamamoto et al. [81] elucidated that HSF1, HSF2, and HSF4 factors in *H. sapiens* can bind to the same HSE, which is characterized by both continuous and discontinuous nGAAn motifs. Notably, these factors exhibit differential recognition and binding to HSEs, indicating a precise interaction mechanism. These findings suggest that the arrangement of nGAAn units in the promoter region plays a crucial role in determining which members of the HSFs are intervening in the regulation of specific genes.

Moreover, the α3-helix within the DBD of EhHSTFs is crucial for interacting with the major groove of HSEs. This interaction primarily involves the serine and arginine residues. This interaction mechanism has been extensively documented in various crystallographic complexes of HSFs, including *K. lactis* (ID: 3HTS) [82], *S. cerevisiae* (ID: 5D5X) [83], and *H. sapiens* (ID: 7DCU) [84], and more recently in *E. histolytica* ([27]. Furthermore, mutations in the amino acids Ser247, Arg250, Asn253, and Tyr255 within the α3-helix of *K. lactis* lead to a lethal phenotype, underscoring the functional importance of these amino acids [82].

The formation of the HSE–HFS complex is a finely regulated process, where diverse molecular interactions play critical roles for its efficiency and precision. In our docking analyses, we found that tEhDBD7 binds to the distinct HSEs of the Ehpgp5, Ehhsp100, Ehmlbp, EhrabB1, and EhrabB1 promoters through hydrogen bonds, hydrophobic interactions, salt bridges, and π–cation interactions (Appendix A). Each type of interaction brings unique characteristics that together ensure effective and specific binding between these elements.

We identified hydrogen bonding interactions between the HSEs and the amino acids Ser69, Arg72, Asn75, and Tyr77, all located in the α3-helix of the tEhDBD7 domain (Appendix A). These amino acids align with the same positions in the α3-helix of *K. lactis*, *S. cerevisiae*, *H. sapiens*, and *E. histolytica*, suggesting a highly conserved interaction pattern in these proteins with HSEs.

In addition to hydrogen bond interactions, hydrophobic interactions also play an essential role in the binding of the complex, providing additional attractive forces between the involved molecules. These interactions contribute to maintaining the correct three-dimensional conformation of the alpha α3-helix in HSF1, as observed in the *H. sapiens* HsHSF1–HSE complex [85,86].

Furthermore, the significance of certain amino acids in the interaction between DBDs and the phosphate group of the HSE backbone through salt bridges has been emphasized [82]. These interactions are key in facilitating the binding of α3-helix to the HSE, contributing significantly to the stability of this complex and the precise regulation of gene expression. They ensure that the DBD of HSF binds to the HSE with high specificity. This mechanism has been observed in the HSF1–HSE complex of *H. sapiens* [84], where amino acids such as Arg71 and Gln72 play a vital role in the formation of salt bridges, thereby increasing the stability of the complex. Furthermore, π–cation interactions are relevant in improving the stability and specificity of ligand–protein complexes, which are formed particularly through the participation of Lys and Arg residues, as reported by Kumar et al. [87]. In our analysis, tEhDBD7 exerts π–cation interactions through the Arg72 residue in all dockings carried out with the HSEs, with the exception of the HSE of the Ehpgp5 promoter.

These results show that the 3D structures of EhDBD5, EhDBD6, and EhDBD7 have a high degree of conservation, consistent with the crystallographic DBDs of *K. lactis*, *S. cerevisiae*, and *H. sapiens*. Furthermore, the interaction mechanism and the specific form in which the α3-helix docks to the main groove of the six HSEs evaluated also show a high degree of conservation. This suggests a highly conserved role in gene regulation through HSFs, recognizing HSEs, and potentially regulating expression efficiently under various conditions faced by the parasite, ensuring its survival. This establishes it as a functional, fundamental, and evolutionarily preserved mechanism in genetic control, not just in *E. histolytica* but across a wide evolutionary spectrum.

Ongoing studies will allow us to better understand the role of EhHSTFs in *E. histolytica* trophozoites, particularly focusing on how these factors influence the behavior of various genes that contain HSEs in their promoters. This research is significant as it aims to elucidate specific pathways in which EhHSTFs control gene activity in different physiological environments. By exploring this topic further, we hope not only to better understand the molecular processes involved but also to potentially identify new targets for therapeutic treatments.

## 4. Materials and Methods

### 4.1. Screening of HSEs in the Promoter Regions of E. histolytica Genes

In a genome-wide screening of HSEs focused on *E. histolytica*, six key HSE sequences identified in gene promoters were considered. These include Ehpgp5 (5′-ataGAAatttTTCata-3′) [28], EhrabB1 (5′-cTTCaTTCcatggtttTTC-3′), EhrabB4 (5′-gTTCttcagtaagaGAA-3′), EhrabB5 (5′-aTTCtacaaaaaGAA-3′) [31], Ehmlbp (5′-agaaGAActaataactGAAaattataatTTC-3′) [30], and Ehhsp100 (5′-aagGAActtGAAgaa-3′) [29]. The promoter regions of 8343 genes in the *E. histolytica* genome, spanning from −500 to +50 base pairs relative to the ATG start codon, were scanned for the six HSE sequences. This search was conducted using the FIMO (Find Individual Motif Occurrences) version 5.5.4 platform (https://meme-suite.org/meme/tools/fimo, accessed on 8 August 2023, Nevada, United States). Sequences with a *p*-value below 1 × 10^−4^ (less than 0.0001) were deemed statistically significant. Subsequently, these identified sequences were entered into the MEME (Multiple Em for Motif Elicitation) platform, version 5.5.4 (MEME Suite), (https://meme-suite.org/meme/tools/meme, accessed on 8 August 2023, Nevada, United States) to create the positional weight matrix (PWM) for each HSE sequence. For the Ehpgp5, Ehhsp100, EhrabB4, and EhrabB5 sequences, two motifs were analyzed, whereas three motifs were examined for the EhrabB1 and Ehmlbp sequences.

### 4.2. Mapping HSEs in Promoter Regions of E. histolytica Genes

The genome sequences and annotations for each HSE were obtained from Amoeba DB v60 (https://amoebadb.org/amoeba/app, accessed on 15 august 2023, Maryland, United States) and the NCBI platform (NCBI) (https://www.ncbi.nlm.nih.gov/, accessed on 15 August 2023, Maryland, United States). The spatial positioning of each HSE within the promoter region, ranging from −500 to +50 bp of each gene, was analyzed using R Studio (Version R 3.6), particularly for creating density plots. Additionally, the presence of three conserved regulatory elements within the gene promoters containing HSEs was investigated. These elements include the atypical TATA box (TATTTAAA); the initiator element Inr (AAAAATTCAA), which overlaps with the transcription start site; and the distinctive amoeba GAAC element (GAACT), as detailed in previous studies by Purdy et al. [35], Vanacova et al. [88], and López-Camarillo et al. [89]. The identification of canonical elements (INR, GAAC, and TATA box) was conducted using the FIMO [90] platform to delineate promoter sequences up to 500 base pairs downstream of the transcription start site. The search was performed using CLC Sequence Viewer software (QIAGEN, 2023, Aarhus, Denmark) (https://digitalinsights.qiagen.com/), documenting the number of elements found in each gene, along with precise details regarding the specific element and its location.

To visualize the conservation of the GAA and TTC motifs, the 1166 sequences identified in the promoter regions of coding genes were inputted into the WebLogo platform (https://weblogo.berkeley.edu/logo.cgi, accessed on 25 August 2023, Berkeyley, United States). This allowed for the generation of a WebLogo, effectively illustrating motif conservation. Additionally, to explore the different combinations of HSEs present within the same promoter, a Venn diagram was constructed. This analysis was facilitated using the online tool available at Bioinformatics PSB UGent Venn Diagram (https://bioinformatics.psb.ugent.be/webtools/Venn/, accessed on 15 September 2023), which provided a clear representation of the overlapping and distinct HSEs.

### 4.3. Gene Ontology Analysis for Genes with HSEs in Their Promoters

The identification of genes was conducted via their accession numbers using the AmoebaDB platform. For gene ontology analysis, the PANTHER 18.0 Released platform (Protein Analysis Through Evolutionary Relationships) was utilized (https://www.pantherdb.org/, accessed on 5 October 2023, California, United States). This comprehensive knowledge base categorizes genes into functional groups such as molecular functions, biological processes, and protein classes. Within each group, there are various functional categories, which are further divided into subcategories to provide a detailed ontological assignment. All the HSEs considered for this ontology analysis possessed a *p*-value less than 1 × 10^−4^ (<0.0001), a threshold deemed statistically significant.

### 4.4. Identification of Genes with HSEs in Reported E. histolytica Transcriptomes

To investigate the relationship between the presence of identified HSEs in the promoters of differentially expressed genes in this study and previously reported transcriptomes of *E. histolytica* trophozoites under various conditions, such as starvation stress [43], intestinal invasion [39], encystation [38], heat shock [49], and normal conditions, we utilized the AmoebaDB platform. After entering the accession numbers of the genes, we downloaded the transcriptomes corresponding to each condition. We then identified the genes containing HSEs uisng their IDs and assessed whether they were upregulated or downregulated in these conditions. This assessment was based on a comparison with genes exhibiting expression levels twice as high or low as those reported in each transcriptome. Furthermore, we analyzed whether these genes harbored single or multiple HSEs and the combinations of these elements.

### 4.5. Three-Dimensional Structural Modeling and Validation of the DNA-Binding Domains (DBDs) of EhHSTF5, EhHSTF6, and EhHSTF7 Factors

To acquire the three-dimensional structures of the DNA-binding domains (EhDBD) of the EhHSTF5, EhHSTF6, and EhHSTF7 factors, the complete protein sequences were retrieved. For this purpose, the respective IDs EHI_137000, EHI_008660, and EHI_200020 from the AmoebaDB platform were used. Amino acids 45 to 137 were used to construct the EhDBD5 domain; for EhDBD6, the segment between amino acid 35 and 125; and for EhDBD7, the region extending between amino acid 17 and 109. These domain-specific ranges were based on predictions made within our team.

The three-dimensional models of the DBDs were generated using the SWISS-MODEL server (https://swissmodel.expasy.org/, accessed on 10 July 2023, Lausana, Suiza). For this process, the crystal structure of the *H. sapiens* HSF1 DNA-binding domain in complex with DNA containing three HSE sites served as a template. This template structure is available in the Protein Data Bank (PDB) (https://www.rcsb.org/structure/7dct, accessed on 10 July 2023 College Station, TX, USA) under the ID 7DCT.

The obtained models were subjected to validation using various servers, including PDBsum (http://www.ebi.ac.uk/thornton-srv/databases/pdbsum/Generate.html, accessed on 19 July 2023, London, United Kingdom), Prosa-Web (https://prosa.services.came.sbg.ac.at/prosa.php, accessed on 30 July 2023, Salzburg, Austria), and UCLA-DOE LAB—SAVES v6.0 (https://saves.mbi.ucla.edu/, accessed on 16 August 2023), which integrates tools such as ERRAT and VERIFY 3D. These tools together allowed for a comprehensive and accurate evaluation of the structural quality of the protein models. The Ramachandran analysis, derived from PDBsum, was used to verify the conformity of the phi and psi angles of the amino acids, ensuring their placement in structurally allowed regions. Prosa-Web offered a comparison with crystallographic structures and assessed the overall energy of the amino acids, thus providing a measure of the model’s stability. ERRAT provided a detailed analysis of structural accuracy through the Global Quality Factor, focusing on non-bonded atomic interactions. Additionally, VERIFY 3D was used to check the congruence between the three-dimensional structure of the model and its amino acid sequence.

### 4.6. Three-Dimensional Structural Modeling of HSEs

For the 3D modeling of the HSEs, Chimera software version 1.17 was utilized. The sequences modeled included 5′-ataGAAatttTTCata-3′ for the HSE of the Ehpgp5 gene, 5′-aagGAActtGAAgaa-3′ for Ehhsp100, 5′-agaaGAActaataactGAAaattataatTTC-3′ for Ehmlbp, 5′-cTTCaTTCcatggtttTTC-3′ for EhrabB1, 5′-gTTCttcagtaagaGAA-3′ for EhrabB4, and 5′-aTTCtacaaaaaGAA-3′ for EhrabB5. Each HSE was modeled in the B-form conformation, which is the typical double-stranded DNA structure. The models were saved in the .pdb file format.

### 4.7. Blind Molecular Dockings and Intermolecular Analysis

Molecular docking analyses were performed using the HDOCK server (http://hdock.phys.hust.edu.cn/, accessed on 21 August 2023). Selection of the best docking option was based on the highest score and highest probability of occurrence. Once docking was completed, the results were downloaded in .pdb format. These results were then meticulously analyzed at the intermolecular level using the Protein-Ligand Interaction Profiler (https://plip-tool.biotec.tu-dresden.de/plip-web/plip/index, accessed on 18 September 2023, Dresden, Germany) server, providing detailed information on protein–ligand interactions.

## 5. Conclusions

This study reveals complex and precise genetic regulation in *E. histolytica* through 2578 HSEs identified in gene promoter regions. These HSEs play a crucial role in regulating genes essential for the organism’s survival and adaptation to various stressors. The potential regulation of these genes by the EhHSTFs underscores the sophistication of genetic mechanisms in *E. histolytica*. Although this discovery is a significant advance, deeper experimental analysis is required to fully understand the role and functioning of these HSEs in the genetic regulation of the parasite.

## Figures and Tables

**Figure 1 ijms-25-01319-f001:**
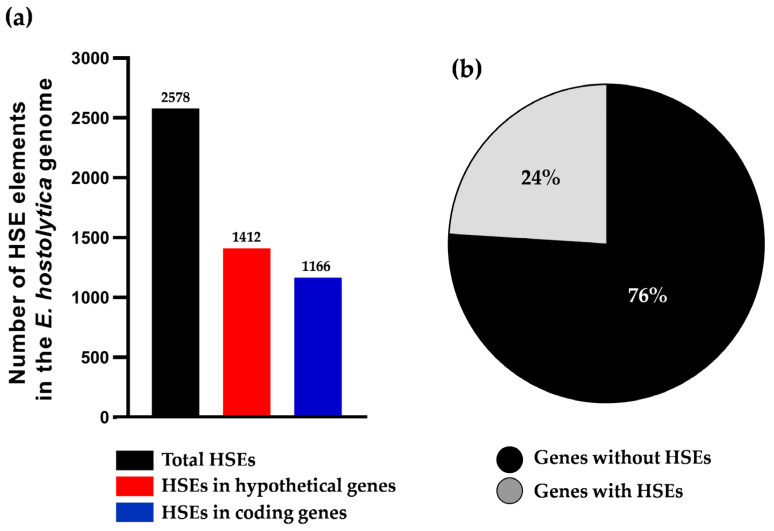
Screening of HSEs in promoter regions of the *E. histolytica* genome. (**a**) Shows the total number of HSEs identified in the genome (black bar), in the promoter regions of hypothetical genes (red bar), and in the promoter regions of coding genes (blue bar). (**b**) Illustrates the percentage of genes in the *E. histolytica* genome that have HSEs in their promoter regions.

**Figure 2 ijms-25-01319-f002:**
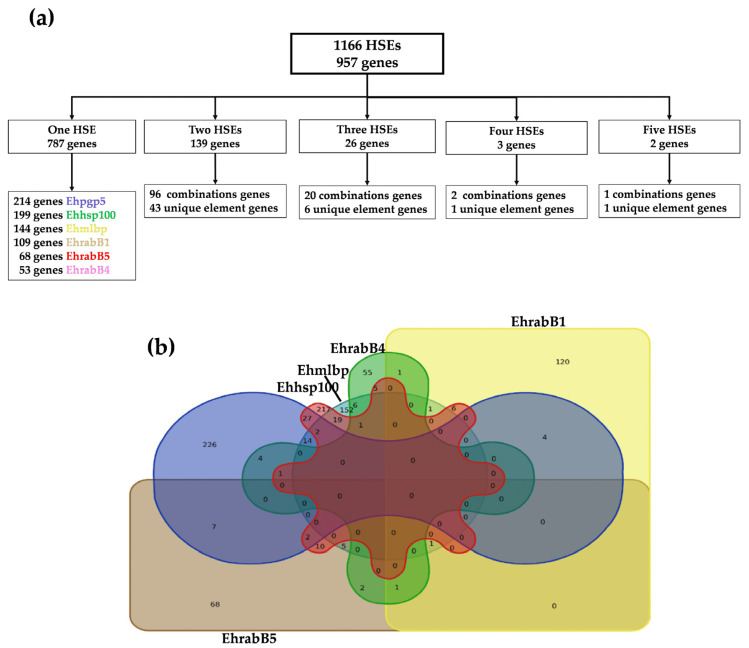
Organization of the six different unique HSE elements or in combination in the gene promoters of *E. histolytica*. (**a**) Number of genes containing HSE elements. (**b**) Detection of combinations of the six HSE elements visualized through a Venn diagram. The elements are color-coded as follows: Ehpgp5 in purple, Ehhsp100 in red, EhrabB4 in green, EhrabB1 in yellow, EhrabB5 in brown, and Ehmlbp in blue.

**Figure 3 ijms-25-01319-f003:**
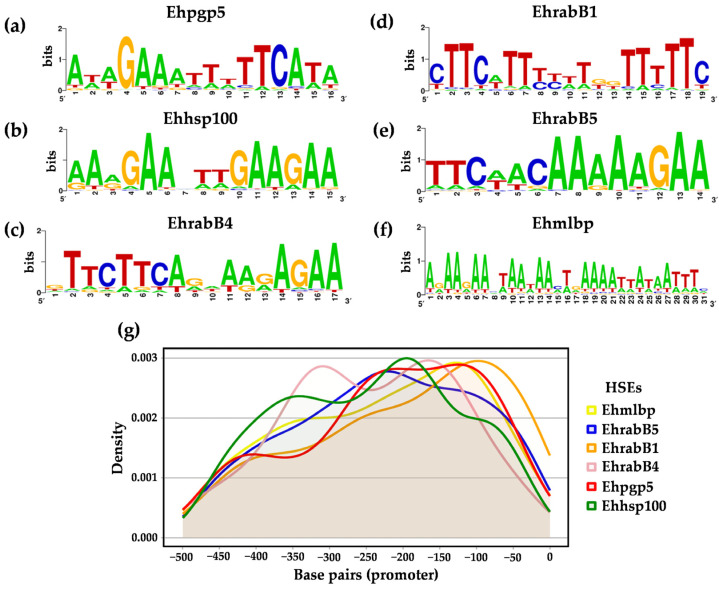
WebLogo sequences and positions of the six HSEs found in the promoters of coding genes in *E. histolytica*. (**a**) Ehpgp5, (**b**) Ehhsp100, (**c**) EhrabB4, (**d**) EhrabB1, (**e**) EhrabB5, and (**f**) Ehmlbp. The WebLogo for each HSE was generated by analyzing the 5′-upstream regions of HSE genes. (**g**) Location of HSEs in the promoter regions (−500 to +50 bp) of the 1166 HSE of *E. histolytica*.

**Figure 4 ijms-25-01319-f004:**
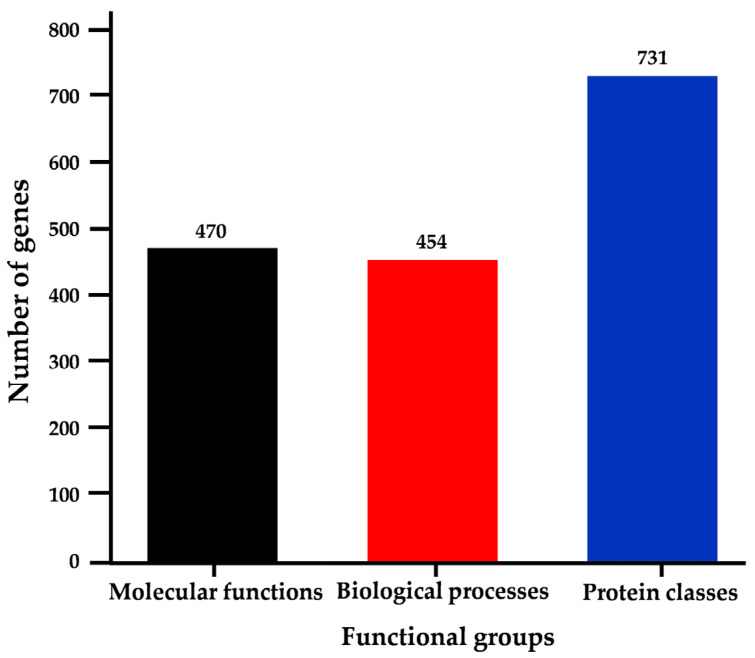
Gene ontology analysis of *E. histolytica*. Categorization of HSE-containing genes into molecular functions, biological processes, and protein classes.

**Figure 5 ijms-25-01319-f005:**
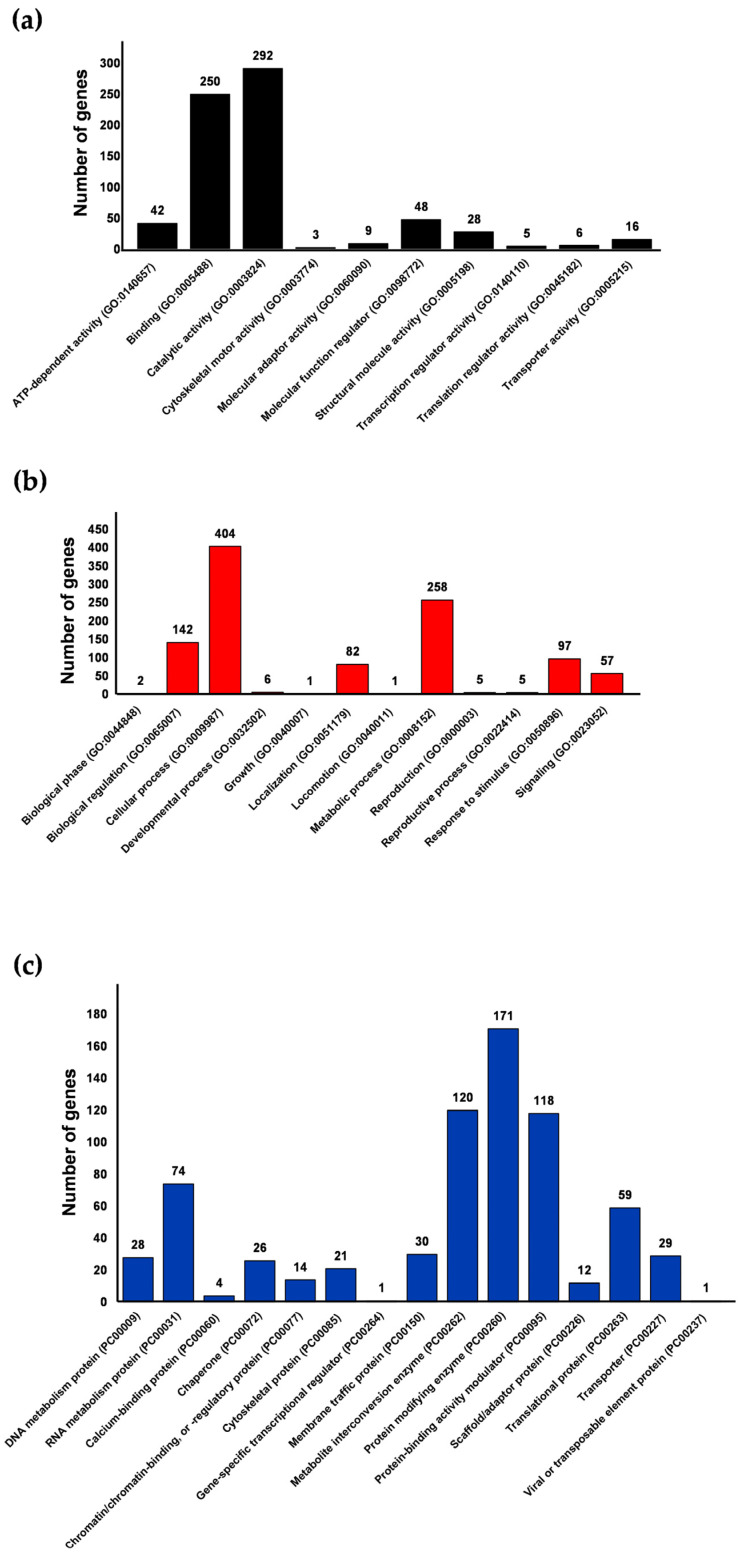
Distribution of HSE-containing genes in *E. histolytica* across ontological categories: (**a**) molecular functions, (**b**) biological processes, and (**c**) protein classes. The entry codes, displayed in parentheses, are identifiers for the categories presented on the PantherDB platform.

**Figure 6 ijms-25-01319-f006:**
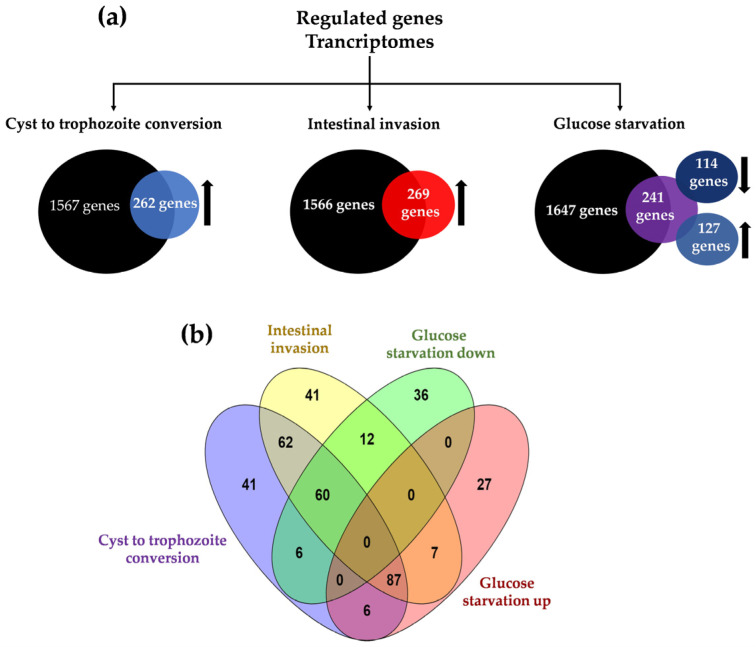
Analysis of HSE-containing genes in *E. histolytica* transcriptomes under cyst-to-trophozoite transition, intestinal invasion, and glucose starvation. (**a**) Overview of reported genes and HSEs presence in each transcriptome. (**b**) Venn diagram of the different transcriptomes; cyst-to-trophozoite conversion (purple), intestinal invasion (yellow), low glucose starvation (green), and high glucose starvation (red). The arrows pointing upwards indicate overexpression, and those pointing downwards indicate underexpression.

**Figure 7 ijms-25-01319-f007:**
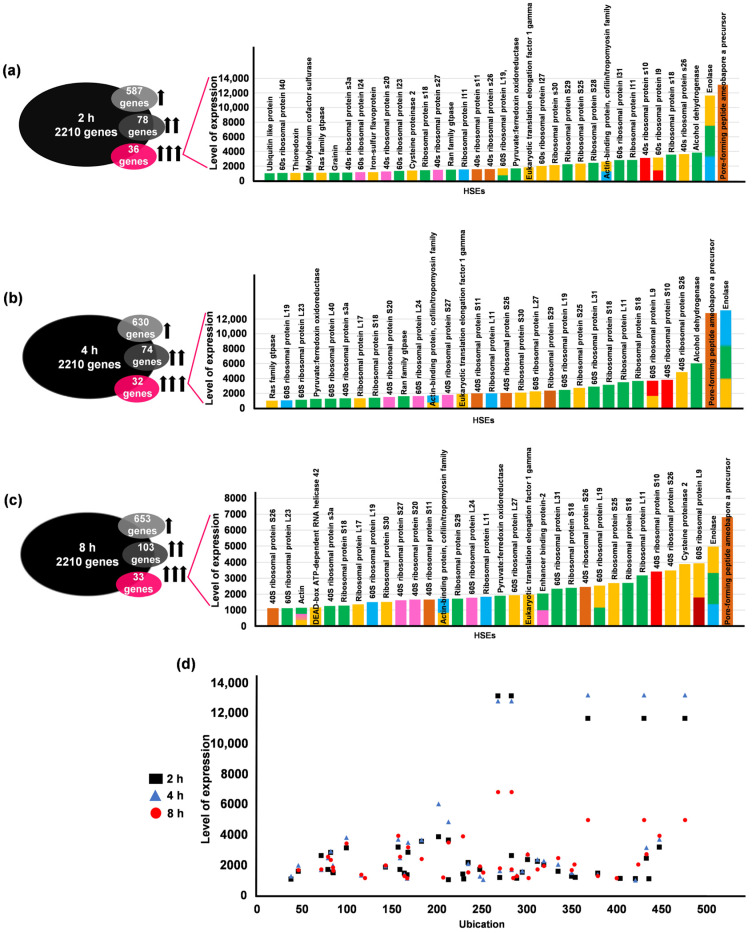
Genes regulated under thermal stress conditions. The analysis was divided into three parts: (**a**) changes in gene expression with HSEs in trophozoites exposed to 2 h, (**b**) 4 h, and (**c**) 8 h of heat shock. Each part included a breakdown of gene expression levels: total genes altering their expression profile (black oval), genes with HSEs and low expression levels (light gray oval), moderate expression levels (dark gray oval), and high expression levels (pink oval). Additionally, each graph highlights the types of HSEs present in the genes within the high-expression group. (**d**) Another analysis was focused on the location of HSEs in the genes with high expression levels at different exposure times: 2 h (black square), 4 h (gray triangle), and 8 h (red circle). The types of HSEs examined were Ehpgp5 (blue bar), Ehhsp100 (green bar), EhrabB4 (pink bar), EhrabB1 (orange bar), EhrabB5 (red bar), and Ehmlbp (yellow bar). One arrow pointing upwards indicates overexpression, two arrows signify significant overexpression, and three arrows represent extreme overexpression.

**Figure 8 ijms-25-01319-f008:**
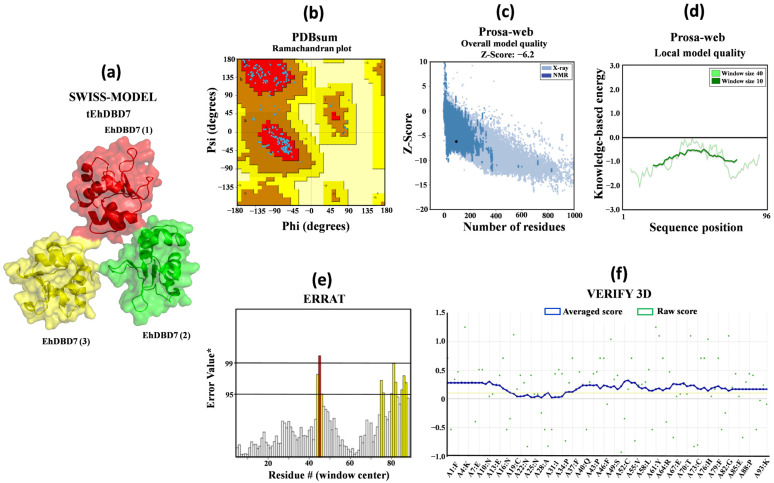
3D structural modeling and validation of tEhDBD7. (**a**) Creation of the tEhDBD7 3D model, followed by exhaustive structural validation using various tools, including (**b**) PDBsum for Ramachandran plot analysis. The letters A, B, and L highlight the Most Favoured Regions. The lowercase a, b, l, and p mark the Additional Allowed Regions, while the prefixed tilde (~a, ~b, ~l, ~p) points to the Generously Allowed Regions. The designation XX is used for the Disallowed Regions. Additionally, a value of -1 is assigned to a torsion angle located in a non-permissible region. (**c**,**d**) Prosa-web was used for the energy profile evaluations, while ERRAT (**e**) was employed for error quantification, with an * denoting the error value. Yellow bars indicate amino acids with structural errors within a margin of error between 95 and 99%, and red bars identify errors exceeding 99%. (**f**) VERIFY 3D for sequence-structure compatibility assessments.

**Figure 9 ijms-25-01319-f009:**
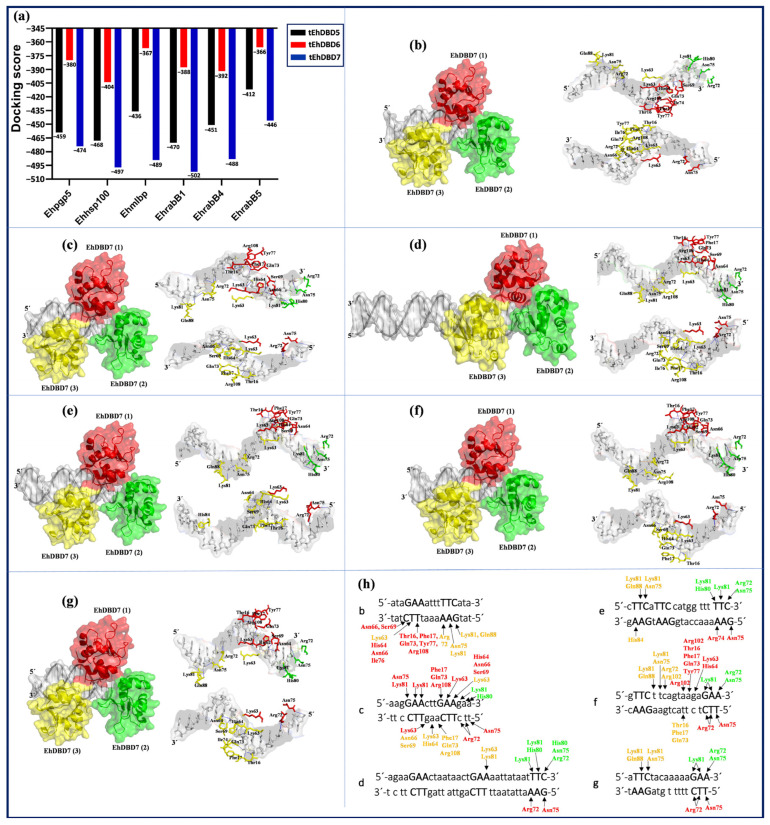
Molecular docking and intermolecular analysis between tEhDBD7 and HSEs. (**a**) The graph shows the binding energy through the docking score of each interaction between various HSEs and tEhDBD5, tEhDBD6, and tEhDBD7. Blind docking was performed between the tEhDBD7 and HSEs: (**b**) Ehpgp5, (**c**) Ehhsp100, (**d**) Ehmlbp, (**e**) EhrabB1, (**f**) EhrabB4, and (**g**) EhrabB5. (**h**) Shows a simplified analysis of the amino acids involved in the recognition of GAA and TTC motifs within the evaluated HSEs. The correspondence between the results is established by matching items in parentheses with the same items without parentheses. For example, ‘(**b**)’ would correspond to ‘b’. To enhance clarity, a color-coding system was used: red represents monomer 1, green for monomer 2, and yellow for monomer 3.

**Table 1 ijms-25-01319-t001:** Total number of HSEs found in the promoter regions of genes in *E. histolytica*.

HSEs	HSE Sequences	Total Sequences	HSE HP	HSE Coding
Ehpgp5	5′-ata**GAA**attt**TTC**ata-3′	645	341	304
Ehhsp100	5′-aag**GAA**ctt**GAA**gaa-3′	702	377	325
EhrabB4	5′-g**TTC**ttcagtaaga**GAA**-3′	163	84	79
EhrabB1	5′-c**TTC**a**TTC**catggttt**TTC**-3′	303	155	148
EhrabB5	5′-a**TTC**tacaaaaa**GAA**-3′	234	136	98
Ehmlbp	5′-agaa**GAA**ctaataact**GAA**aattataat**TTC**-3′	531	319	212

The capital letters highlighted in bold are the motifs contained in the HSEs (GAA and TTC), which are recognized by the EhHSTFs for joining. HP, hypothetic. The *p*-value of all sequences is less than 0.0001.

**Table 2 ijms-25-01319-t002:** Number of genes identified for each HSE.

HSEs	Number of Genes					Total
	One HSE	Two HSEs	Three HSEs	Four HSEs	Five HSEs	
Ehpgp5	214	59	11	2	1	287
Ehhsp100	199	70	16	3	2	290
EhrabB4	53	19	4	0	0	76
EhrabB1	109	20	5	0	0	134
EhrabB5	68	20	8	0	0	96
Ehmlbp	144	47	10	0	0	201

The presence of two to five HSEs in genes can result from either a single array or a combination of multiple HSEs.

**Table 3 ijms-25-01319-t003:** Regulatory elements including the TATA, Inr, and GAAC box in HSE promoters in *E. histolytica*.

	Number of Genes						
Regulatory Elements		Ehmlbp	Ehpgp5	Ehhsp100	EhrabB1	EhrabB4	EhrabB5
TATA	68	14	17	25	13	8	4
GAAC	298	59	94	104	42	20	26
Inr	3	0	0	0	1	1	1
TATA/GAAC	24	5	10	11	6	1	2
TATA/Inr	1	0	0	0	0	0	1
GAAC/Inr	0	0	0	0	0	0	0
TATA/GAAC/Inr	0	0	0	0	0	0	0
The promoters of *E. histolytica* have different HSE numbers and regulatory elements							
Regulatory Elements	Number of Genes	1 HSE	2 HSE	3 HSE			
TATA	68	56	10	2			
GAAC	298	254	41	3			
INR	3	3	0	0			

**Table 4 ijms-25-01319-t004:** Genes with HSE elements are expressed during cyst-to-trophozoite conversion, intestinal invasion, glucose starvation and exposed to heat stress.

Stress Condition	Number of Genes					Total
	One HSE	Two HSE	Three HSE	Four HSE	Five HSE	
Cyst-to-trophozoite						
	220	31	9	2	-	262
Intestinal colonization						
	225	34	8	2	-	269
Glucose starvation down						
	94	15	5	0	-	114
Glucose starvation up						
	113	9	4	1	-	127
Heat shock (2 h)						
No expression	208	42	6			256
Low	486	80	17	2	2	587
Medium	63	12	2	1		78
High	30	5	1			36
Heat shock (4 h)						
No expression	183	34	4			221
Low	517	90	19	2	2	630
Medium	61	10	2	1		74
High	26	5	1			32
Heat shock (8 h)						
No expression	134	30	4			168
Low	538	93	18	2	2	653
Medium	90	10	2	1		103
High	26	6	2			33

**Table 5 ijms-25-01319-t005:** The most frequent HSEs under different stress conditions.

Stress Condition	HSE Most Common (HSE Gene Number)				Second HSE Most Common (HSE Gene Number)			
Cyst-to-trophozoite	Ehhsp100 (86)				Ehpgp5 (79)			
Intestinal colonization	Ehhsp100 (102)				Ehpgp5 (69)			
Glucose starvation down	Ehhsp100 (54)				Ehpgp5 (31)			
Glucose starvation up	Ehmlbp (44)				Ehhsp100 (32)			
Heat shock	No expression	Low	Medium	High	No expression	Low	Medium	High
2 h	Ehhsp100 (89)	Ehpgp5 (193)	Ehhsp100 (35)	Ehhsp100 (18)	Ehpgp5 (83)	Ehhsp100 (182)	Ehpgp5 (25)	Ehmlbp (13)
4 h	Ehpgp5 (75) and Ehhsp100 (75)	Ehpgp5 (205)	Ehhsp100 (33)	Ehhsp100 (15)	EhrabB1 (43)	Ehhsp100 (201)	Ehmlbp (25)	Ehmlbp (11)
8 h	Ehhsp100 (60)	Ehpgp5 (213)	Ehhsp100 (46)	Ehhsp100 (14)	Ehpgp5 (57)	Ehhsp100 (204)	Ehpgp5 (30)	Ehmlbp (13)

## Data Availability

The datasets used and/or analyzed during the current study are available from the corresponding author upon reasonable request.

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
