# Peer review of "Screening and Structural Characterization of Heat Shock Response Elements (HSEs) in Entamoeba histolytica Promoters"

_ijms, 2024, doi:10.3390/ijms25021319_

Round 1

Reviewer 1 Report

Comments and Suggestions for Authors

Dear Authors,

Please check an attached file for my comments.

Best regards,

Reviewer 2 Report

Comments and Suggestions for Authors

This manuscript is a very comprehensive study of the heat shock response elements in the E. histolytica genome. The study is significant in increasing our understanding of the complex survival mechanisms of the parasite. The study is very well done, and I have only a few comments below

L38 full genus name (Entamoeba) must be written ant the beginning of the sentence

L108-111 these sentences are more suitable for conclusions. delete or modify as appropriate to the research objectives (L105)

L184-185 Figure 2 caption is missing

L123, 126, 239, 241, 312,484,573 species name must be in italic

L303-304 it is slightly confusing presentation of data, three distinguished groups overlaps and exceeds 100%, this should be reflected in the figure.

L451,480 for biological species avoid such grammatical forms as “E. histolytica's”
